# The effects of genetic and modifiable risk factors on brain regions vulnerable to ageing and disease

Jordi Manuello [1,2], Joosung Min [3], Paul McCarthy[1], Fidel Alfaro-Almagro[1], Soojin Lee[1,4], Stephen Smith[1], Lloyd T. Elliott[3,7], Anderson M. Winkler[5,6,7] & Gwenaëlle Douaud [1] ✉

We have previously identified a network of higher-order brain regions particularly vulnerable to the ageing process, schizophrenia and Alzheimer's disease. However, it remains unknown what the genetic influences on this fragile brain network are, and whether it can be altered by the most common modifiable risk factors for dementia. Here, in ~40,000 UK Biobank participants, we first show significant genome-wide associations between this brain network and seven genetic clusters implicated in cardiovascular deaths, schizophrenia, Alzheimer's and Parkinson's disease, and with the two antigens of the XG blood group located in the pseudoautosomal region of the sex chromosomes. We further reveal that the most deleterious modifiable risk factors for this vulnerable brain network are diabetes, nitrogen dioxide – a proxy for traffic-related air pollution – and alcohol intake frequency. The extent of these associations was uncovered by examining these modifiable risk factors in a single model to assess the unique contribution of each on the vulnerable brain network, above and beyond the dominating effects of age and sex. These results provide a comprehensive picture of the role played by genetic and modifiable risk factors on these fragile parts of the brain.

The development of preventative strategies based on modifying risk factors might prove to be a successful approach in ensuring healthy ageing. Factors particularly scrutinised in dementia and unhealthy ageing have included cerebrovascular factors such as high blood pressure, diabetes and obesity, but also lifestyle ones such as alcohol consumption, and protective factors such as exercise[1]. Assessing these modifiable risk factors together makes it possible to identify the unique contribution of each of these factors on the brain or on cognitive decline. A Lancet commission, updated in 2020 to include, e.g., pollution for its possible role in the incidence of dementia[2], examined the relative impact of 12 modifiable risk factors for dementia, and showed that these 12 factors may account for 40% of the cases worldwide[3]. Conversely, genetic factors are non-modifiable in nature, but can inform us about the mechanisms underlying the phenotypes of interest. These mechanisms sometimes can be shared across these

phenotypes. For instance, genetic overlap has been found for Alzheimer's and Parkinson's diseases at a locus in the *MAPT* region[4]. Likewise, one of the most pleiotropic variants, in the *SLC39A8/ZIP8* gene, shows genome-wide associations with both schizophrenia and fluid intelligence, amongst many other phenotypes[5,6].

One way to objectively and robustly assess susceptibility for unhealthy ageing is to look non-invasively at brain imaging markers[7]. Using a data-driven approach on a lifespan cohort, we previously identified an ensemble of higher-order, 'transmodal' brain regions that degenerates earlier and faster than the rest of the brain[8]. The very same areas also develop relatively late during adolescence, thus supporting the 'last in, first out' (LIFO) hypothesis, which posits that the process of age-related brain decline mirrors developmental maturation. Importantly, this network of brain regions further demonstrated heightened vulnerability to schizophrenia and Alzheimer's disease, two disorders

A full list of author affiliations appears at the end of the paper. ✉e-mail: gwenaelle.douaud@ndcn.ox.ac.uk

that impact on brain structure during adolescence and ageing respectively. Accordingly, this LIFO network was strongly associated with cognitive traits whose impairment is specifically related to these two disorders, namely fluid intelligence and long-term memory[8].

Here, our main objective was to assess both the genetic and modifiable risk factors' contributions to the vulnerability of these most fragile parts of the brain. We conducted a genome-wide association study on a prospective cohort of nearly 40,000 participants of the UK Biobank study who had received brain imaging, and in total evaluated the association between the LIFO brain network and 161 modifiable risk factors, classified according to 15 broad categories: blood pressure, cholesterol, diabetes, weight, alcohol consumption, smoking, depressive mood, inflammation, pollution, hearing, sleep, socialisation, diet, physical activity and education.

## Results

### The vulnerable LIFO brain network in UK Biobank

Similar to our previously observed results[8], the loadings of the LIFO brain network, i.e., the normalised grey matter volume in the network after regressing out the effects of all the other brain maps (see Methods), demonstrated a strong quadratic association with age in the UK Biobank cohort of 39,676 participants ($R^2 = 0.30$, $P < 2.23 \times 10^{-308}$,

Fig. 1). These higher-order regions thus show an accelerated decrease of grey matter volume compared with the rest of the brain. Furthermore, these areas define a network mainly involved in behavioural tasks related to execution, working memory, and attention (Fig. 1, Supplementary Information).

### Genetic influences over the vulnerable LIFO brain network

Using a minor allele frequency filter of 1% and a $-\log_{10}(P)$ threshold of 7.5, we found, in the 39,676 participants, genome-wide associations between the LIFO brain network and seven genetic clusters whose top variants were all replicated (Table 1/Supplementary Data 1, Fig. 2).

The first autosomal genetic cluster, on chromosome 1, included two variants (lead variant: rs6540873, $\beta = 0.06$, $P = 1.71 \times 10^{-8}$, and rs1452628, with posterior probabilities of inclusion in the causal variant set of 0.56 and 0.45, respectively) close to, and eQTL of, *KCNK2* (*TREK1*). This gene regulates immune-cell trafficking into the central nervous system, controls inflammation, and plays a major role in the neuroprotection against ischemia. Of relevance, these two loci are in particular related in UK Biobank participants with the amount of alcohol consumed, insulin levels, inflammation with interleukin-8 levels, as well as, crucially, with late-onset Alzheimer's disease (Table 1/Supplementary Data 1).

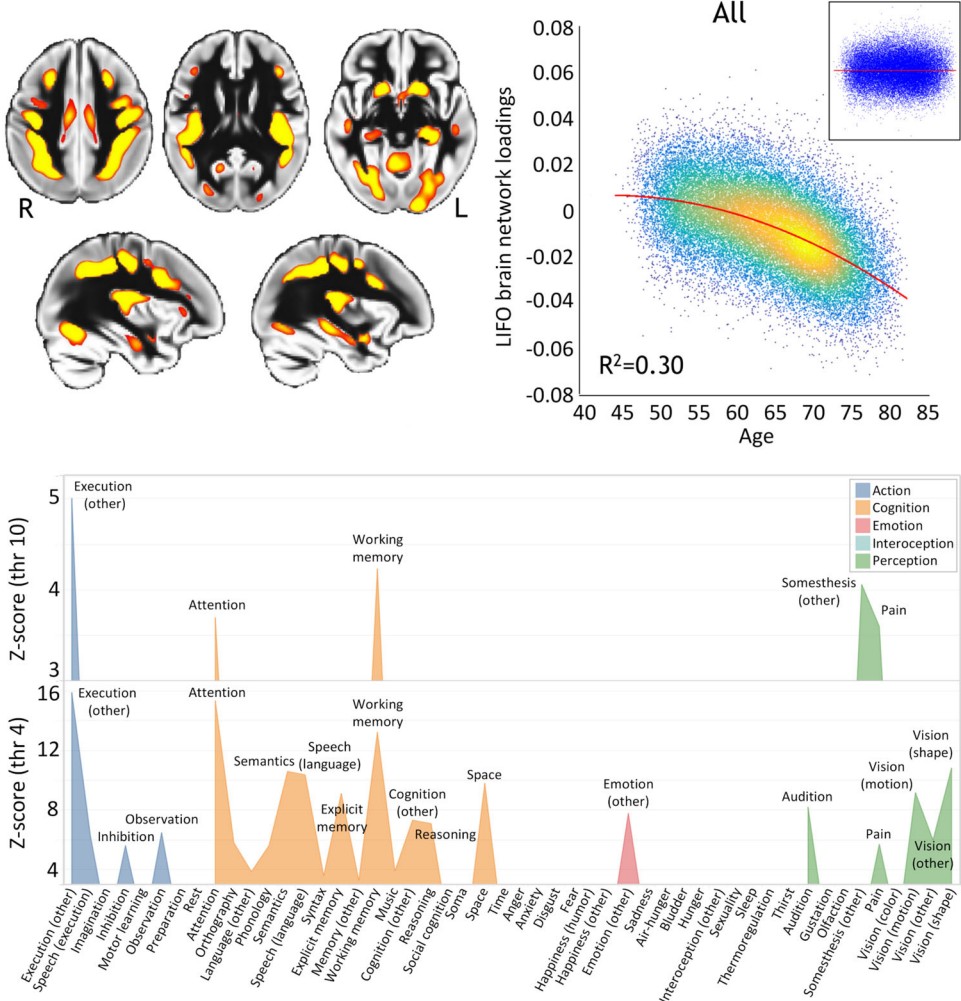

**Fig. 1 | Vulnerable 'last in, first out' (LIFO) network of higher-order brain regions that degenerate earlier and faster than the rest of the brain.** Top left, spatial map of the LIFO network (in red-yellow, thresholded at $Z > 4$ for visualisation) used to extract the loadings from every scanned participant from UK Biobank ($n = 39,676$). Top right, these LIFO loadings (in arbitrary units) show a strong quadratic association with age in the UK Biobank cohort, i.e. grey matter volume decreases quadratically with older age in these specific regions ($R^2 = 0.30$, $P < 2.23 \times 10^{-308}$; inset: residual scatterplot). Bottom, the vulnerable network appears to encompass areas mainly involved in execution, working memory, and attention (using the BrainMap taxonomy[60], and with the LIFO brain network thresholded at both $Z = 4$ and $Z = 10$, see Supplementary Information).

**Table 1 | Summary of significantly associated genetic clusters with the vulnerable 'last in, first out' brain network**

| | Significant variants | N | P-value | beta | Chr | Position | a1 | a2 | af | Gene | eQTL, tissue | Reproduction P-value |
|---|---|---|---|---|---|---|---|---|---|---|---|---|
| 1 | **rs6540873** rs1452628 | 2 | 1.71E−08 | 0.06 | 1 | 215,137,222 | A | C | 0.38 | intergenic, ~40 kb from *KCNK2 (TREK1)* | *KCNK2 (TREK1)* Non-brain | 3.92E−04 |
| 2 | **rs13107325** rs13135092 rs35518360 rs35225200 rs13105682 rs6855246 rs1813006 | 7 | 2.82E−13 | 0.14 | 4 | 103,188,709 | C | T | 0.07 | exon (missense), *SLC39A8 (ZIP8)* | *UBE2D3* Non-brain | 1.18E−03 |
| 3 | **rs35187443** (6:45442860_TA_T) | 1 | 9.03E−09 | 0.06 | 6 | 45,442,860 | TA | T | 0.38 | intron of *RUNX2* | in high LD ($R^2$ = 0.86) with rs2677109, eQTL of *RUNX2* Brain | 3.05E−03 |
| 4 | **rs12146713** | 1 | 1.26E−09 | −0.10 | 12 | 106,476,805 | T | C | 0.09 | intron of *NUAK1 (ARK5)* | RP11-114F10.2 Non-brain | 5.64E−03 |
| 5 | **rs2532395** (highest hit after tri-allelic rs2693333)[a] | 3,906 | 3.56E−15 | −0.09 | 17 | 44,307,193 | C | T | 0.21 | intergenic, in *MAPT* region, <10 kb from *KANSL1* | multiple genes, including *MAPT, KANSL1, CRHR1, SPPL2C, ARL17A* Non-brain | 1.97E−02 |
| 6 | **rs312238** rs312241 rs312245 rs312250 rs113075535 rs312257 rs312255 rs312256 | 8 | 1.77E−10 | −0.05 | X Y | 2,669,044 2,619,044 | G | T | 0.15 | intergenic, ~1 kb from *XG*, ~10 kb from *CD99* | *CD99* Non-brain | 1.12E−06 |
| 7 | **rs2857316** rs2109378 rs2534628 rs2534630 rs2534629 rs311155 rs28620378 rs28758440 rs2534635 | 9 | 2.27E−29 | −0.08 | X Y | 2,698,954 2,648,954 | G | A | 0.32 | intron of *XG* intron of *XGPY2* | *XG* Non-brain | 3.87E−19 |

Lead variant in bold. Due to formatting constraints, this is an abbreviated version without e.g., the association with nIDPs; the full table is available as Supplementary Data 1. *P*-values are derived from a two-sided linear association test.
[a]Full list of significant hits in cluster in Supplementary Data 4.

The second autosomal genetic cluster on chromosome 4 was made of 7 loci, with the lead variant rs13107325 in an exon of *SLC39A8/ZIP8* ($\beta$ = 0.14, P = 2.82 × 10$^{-13}$, posterior probability: 0.99). This locus is one of the most pleiotropic SNPs identified in GWAS, and is, amongst many other associations, related in UK Biobank with cholesterol, blood pressure, weight, inflammation with C-reactive proteins levels, diabetes with insuline-like growth factor 1 levels, alcohol intake, sleep duration, and cognitive performance/impairment, including prospective memory (Table 1/Supplementary Data 1).

The third locus was an indel in chromosome 6 in an intron, and eQTL, of *RUNX2* (rs35187443, $\beta$ = 0.06, P = 9.03 × 10$^{-9}$), which plays a key role in differentiating osteoblasts, and has been very recently shown to limit neurogenesis and oligodendrogenesis in a cellular model of Alzheimer's disease[9].

The fourth locus was a SNP in chromosome 12, in an intron of *NUAK1* (rs12146713, $\beta$ = −0.10, P = 1.26 × 10$^{-9}$), and remarkably its top association in UK Biobank was with the contrast between schizophrenia and major depressive disorder[10], and it was also associated with insulin-like growth factor 1 levels (Table 1/Supplementary Data 1).

The final genetic autosomal genetic cluster was made of 3,906 variants in the *MAPT* region. Its lead non-triallelic variant, rs2532395 ($\beta$ = −0.09, P = 3.56 × 10$^{-15}$) was more specifically <10 kb from *KANSL1* and an eQTL of *KANSL1*, *MAPT* and other genes in brain tissues (Table 1/Supplementary Data 1, Supplementary Data 4). This locus was also associated in UK Biobank with tiredness and alcohol intake. *MAPT* is in

17q21.31, a chromosomal band involved with a common chromosome 17 inversion[11]. Adding chromosome 17 inversion status as a confounder reduced the significance of the association ($\beta$ = −0.15, P = 8.45 × 10$^{-3}$). Since the genotype for rs2532395 was also strongly correlated with chromosome 17 inversion in our dataset (Pearson correlation $r$ = 0.98, $P < 2 \times 10^{-16}$), this would suggest that the association between *MAPT* and the LIFO network is not independent from chromosome 17 inversion. As this extended genetic region is known for its pathological association with many neurodegenerative disorders including Alzheimer's disease, we investigated whether the LIFO brain regions mediated the effect of the *MAPT* genetic cluster (using the lead bi-allelic variant rs2532395) on Alzheimer's disease (see Methods). Despite small average causal mediated effect (ACME) sizes, we found a significant effect for both the dominant model (ACME $\beta$ = 1.16 × 10$^{-4}$; 95% CI = [5.19 × 10$^{-5}$, 1.99 × 10$^{-4}$]; P = 4 × 10$^{-5}$) and the recessive model (ACME $\beta$ = 1.55 × 10$^{-4}$; 95% CI = [3.96 × 10$^{-5}$, 3.74 × 10$^{-4}$]; P = 4 × 10$^{-5}$; full output of the mediation package on the dominant and recessive models in Supplementary Information).

The two last genetic clusters of 8 and 9 variants respectively were found on the X chromosome, notably in a pseudo-autosomal region (PAR1), which is interestingly hit at a higher rate than the rest of the genome ($P = 1.56 \times 10^{-5}$, see Supplementary Information). The top variants for these clusters were related to two homologous genes coding for the two antigens of the XG blood group: rs312238 ($\beta$ = −0.05, P = 1.77 × 10$^{-10}$) ~10 kb from, and an eQTL of, *CD99/MIC2*,

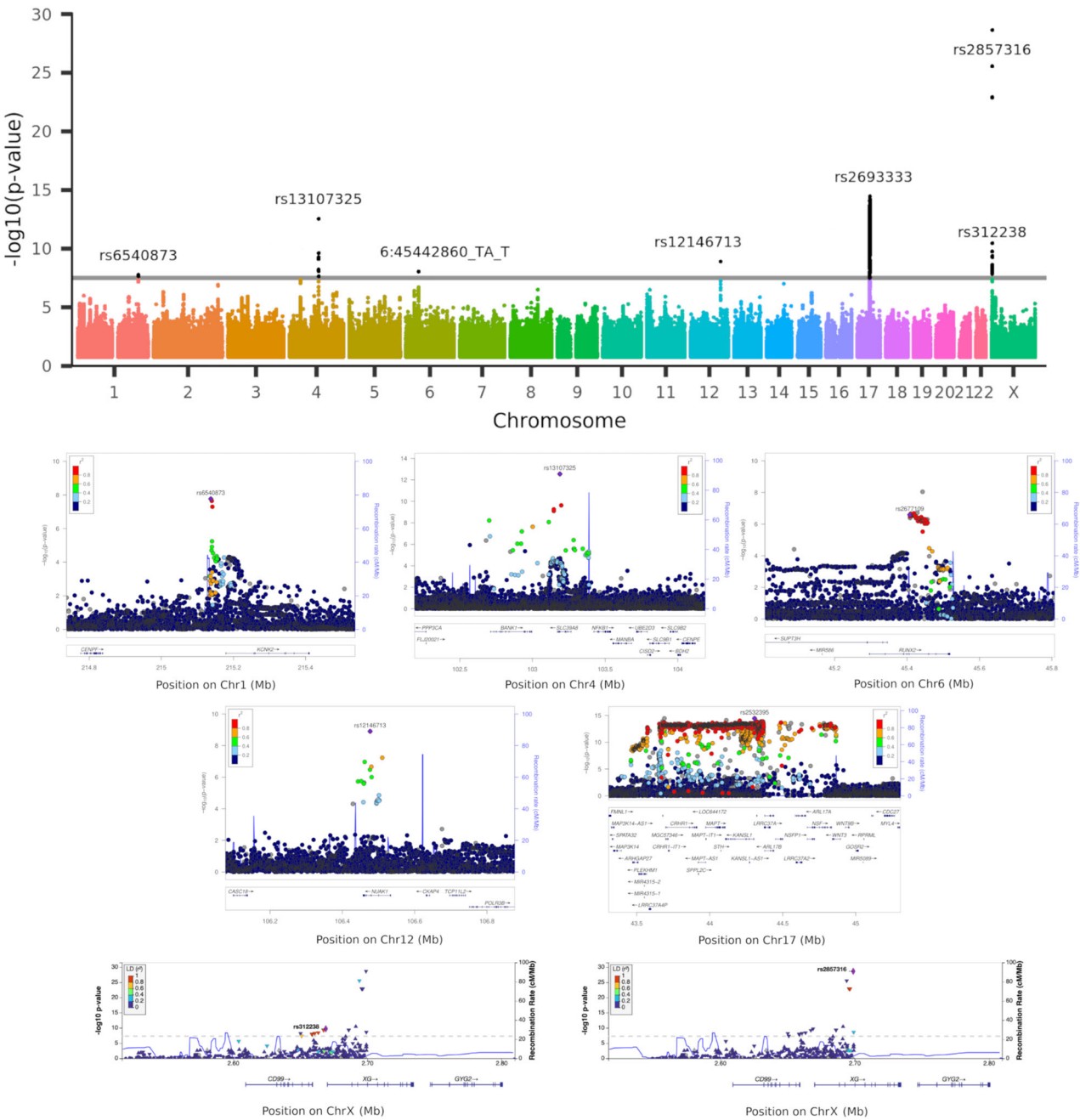

**Fig. 2 | Manhattan plot and regional autosomal association plots for the variants significantly associated genome-wide with the vulnerable 'last in, first out' (LIFO) brain network.** Top row, Manhattan plot showing the 7 significant genetic clusters associated with the LIFO brain network ($-\log_{10}(P) > 7.5$). Second and third rows, regional association plots of the top variants for each of the 5 autosomal genetic clusters: rs6540873 on chromosome (Chr) 1 (*KCNK2*), rs13107325 on Chr4 (*SLC39A8*), rs2677109 on Chr6 (*RUNX2*) (as a proxy in high LD $R^2 = 0.86$ with indel 6:45442860_TA_T), rs12146713 on Chr12 (*NUAK1*), and rs2532395 on Chr17 (*MAPT, KANSL1*)(highest variant after tri-allelic rs2693333; see Supplementary Data 4 for a complete list of significant variants in this 5th *MAPT* genetic cluster). Bottom row, regional association plots of the top variants for the two genetic clusters in the pseudo-autosomal region PAR1 of the X chromosome: rs312238 (*XG, CD99*) and rs2857316 (*XG*)(UK Biobank has no genotyped variants on the 3' side). Based on Human Genome build hg19. *P*-values are derived from a two-sided linear association test.

and rs2857316 ($\beta = -0.08$, $P = 2.27 \times 10^{-29}$) in an intron and eQTL of *XG* (Table 1/Supplementary Data 1). Since chromosome X has hardly been explored, we carried out our own association analyses between these two top variants and non-imaging variables in UK Biobank. Intriguingly, the first of these two PAR1 loci, rs312238, was found to be significantly associated in the genotyped participants who had not been scanned (out-of-sample analysis in $n = 374,230$ UK Biobank participants) with nitrogen dioxide air pollution, our 'best' MRF for pollution (see below), and many other environmental, socioeconomic, and early life factors (such as urban or rural setting, distance from the coast, place of birth, number of siblings, breastfed as a baby, maternal smoking around birth), as well as health outcomes (Supplementary Data 2). In particular, amongst the more easily interpretable findings of the most associated variables with rs312238, the T allele of this locus was associated with two increased measures of deprivation and/or disability (worse socioeconomic status), the 'Townsend deprivation index' and the 'Health score', but also with 'Nitrogen dioxide air pollution', 'Maternal smoking around birth', as well as 'Number of full

**Table 2 | Details and associations with the vulnerable 'last in, first out' brain network for each modifiable risk factor (MRF) across 15 different categories**

| Category | Visit | MRFs description | With confounders | | | Without confounders | | | |
|---|---|---|---|---|---|---|---|---|---|
| | | | $R^2$ | r | P-value | $R^2$ | r | P-value | n |
| Alcohol consumption | 2 | Alcohol intake frequency | 0.002 | −0.04 | 8.53E−18 | 0.009 | −0.10 | 1.35E−80 | 39,398 |
| Body size measurements | 0 | Waist circumference | 0.002 | −0.05 | 5.97E−22 | 0.039 | −0.20 | <2.23−308 | 39,644 |
| Blood pressure | 2 | Medication for blood pressure | 0.002 | −0.05 | 7.55E−22 | 0.032 | −0.18 | 5.43E−273 | 37,922 |
| Cholesterol | 2 | Medication for cholesterol | 0.002 | −0.04 | 2.20E−14 | 0.051 | −0.22 | <2.23−308 | 37,922 |
| Diabetes | 2 | Diabetes diagnosed by doctor | 0.005 | −0.07 | 1.51E−41 | 0.011 | −0.10 | 3.43E−96 | 39,306 |
| Exercise | 2 | Frequency of stair climbing in last 4 weeks | 0.001 | 0.03 | 8.91E−08 | 0.003 | 0.05 | 1.07E−24 | 39,372 |
| Hearing | 0 | Hearing difficulty/background | 0.000 | −0.02 | 1.36E−04 | 0.010 | −0.10 | 6.05E−86 | 38,970 |
| Inflammation | 2 | Use of medication for pain relief | 0.000 | −0.02 | 1.15E−05 | 0.001 | −0.03 | 1.92E−10 | 39,132 |
| Pollution | a | Nitrogen dioxide air pollution in 2005 | 0.003 | −0.05 | 2.75E−24 | 0.000 | −0.01 | 2.22E−02 | 39,211 |
| Sleep | 2 | Sleep duration | 0.001 | −0.03 | 5.20E−09 | 0.006 | −0.08 | 2.53E−54 | 39,313 |
| Smoking | 2 | Past tobacco smoking | 0.002 | −0.05 | 9.53E−19 | 0.012 | −0.11 | 3.37E−105 | 38,481 |
| Socialisation | 0 | Pub or social clubs | 0.000 | −0.02 | 2.07E−05 | 0.002 | −0.04 | 7.99E−18 | 39,609 |

Due to formatting constraints, this is an abbreviated version presenting only the 'best' MRFs; the full table is available as Supplementary Data 3. *P*-values are derived from a two-sided linear association test.
ᵃUpdated regularly.

**Table 3 | Single model comprising the 12 'best' modifiable risk factors (MRFs) and 6 confounders showing associations with the vulnerable 'last in, first out' (LIFO) brain network**

| Whole model: selected 'best' MRFs and confounders | | | | | | |
|---|---|---|---|---|---|---|
| Category | MRFs description | $R^2$ | r | p-value | n | Model $R^2$ |
| **Diabetes** | **Diabetes diagnosed by doctor** | **0.003** | **−0.054** | **1.13E−24** | **35,527** | 0.0145 |
| **Pollution** | **Nitrogen dioxide air pollution in 2005** | **0.002** | **−0.049** | **5.39E−20** | **35,527** | |
| **Alcohol** | **Alcohol intake frequency** | **0.002** | **−0.045** | **3.81E−17** | **35,527** | |
| Sleep | Sleep duration | 0.001 | −0.028 | 1.39E−07 | 35,527 | |
| Body size measurements | Waist circumference | 0.001 | −0.027 | 2.99E−07 | 35,527 | |
| Smoking | Past tobacco smoking | 0.001 | −0.027 | 3.14E−07 | 35,527 | |
| Blood pressure | Medication for blood pressure | 0.001 | −0.025 | 1.61E−06 | 35,527 | |
| Exercise | Frequency of stairs climbing in last 4 weeks | 0.000 | 0.020 | 2.34E−04 | 35,527 | |
| Hearing | Hearing difficulty/Background | 0.000 | −0.014 | 1.09E−02 | 35,527 | |
| Inflammation | Use of medication for pain relief | 0.000 | −0.010 | 5.12E−02 | 35,527 | |
| Socialisation | Pub or social clubs | 0.000 | −0.007 | 1.83E−01 | 35,527 | |
| Cholesterol | Medication for cholesterol | 0.000 | −0.007 | 1.97E−01 | 35,527 | |

'Best' MRFs are ranked according to their |r| values. Significant results after conservative correction for multiple comparisons are in bold (see Supplementary Information - Modifiable risk factors two-stage analysis). *P*-values are derived from two-sided linear association tests.

brothers' and 'Number of full sisters', thus showing consistent signs of association between this variant and these phenotypes.

We found that the heritability of the LIFO network was significant, with $h^2 = 0.15$ (se = 0.01). The genetic co-heritability between the LIFO network and Alzheimer's disease or schizophrenia was not statistically significant (coefficient of co-heritability = −0.12, se = 0.10; $P = 0.23$; coefficient of co-heritability = −0.16, se = 0.04, $P = 0.07$, respectively).

**Modifiable risk factors' associations with the vulnerable LIFO brain network**

Including the modifiable risk factors (MRFs) in a single general linear model allows us to assess the unique contribution of each factor on the LIFO brain network. Not all UK Biobank participants have data available for all of the MRF variables however. An analysis limited to those with complete data for all MRFs would be biased, and based on a relatively small, low-powered sample. We addressed this issue via a two-stage analysis in which: (i) we first identified which variable within each of the 15 MRF categories best represented associations of that category with

the LIFO brain network loadings (based on two criteria: significance and <5% missing values), (ii) we investigated the unique contribution of that MRF category, over and above all other categories and the dominating effects of age and sex, to the LIFO loadings.

From the first stage of our analysis, 12 of the 15 categories of MRFs had at least one 'best' MRF, i.e., with a significant effect on the LIFO brain network and enough non-missing values across all scanned participants to be investigated further (Table 2/Supplementary Data 3). The contribution of the MRFs on the vulnerable brain network differed vastly depending on whether confounding effects of age, sex and head size were taken into account. The effect size and significance of some MRFs diminished because of some clear collinearity with the confounders. For instance, for the category of blood pressure, the most significant MRF was first "systolic blood pressure, automatic (second) reading" ($r = −0.20$, $P < 2.23 \times 10^{-308}$), but after regressing out the confounders, the 'best' MRF for this category was "medication for blood pressure" ($r = −0.05$, $P = 7.55 \times 10^{-22}$). Conversely, regressing out the effects of age served to unmask the significant deleterious effects

of pollution on the vulnerable brain regions, such as nitrogen dioxide air pollution or particulate matter air pollution (Table 2/Supplementary Data 3).

When considered together in a single model in the second stage of the analysis, 3 best MRFs had an effect on the LIFO brain network that remained significant beyond the dominating effects of age and sex, and of the 9 other best MRFs: diabetes ("diabetes diagnosed by doctor", $r = -0.05$, $P = 1.13 \times 10^{-24}$), pollution ("nitrogen dioxide air pollution in 2005", $r = -0.05$, $P = 5.39 \times 10^{-20}$) and alcohol ("alcohol intake frequency", $r = -0.04$, $P = 3.81 \times 10^{-17}$) (Table 3). No MRFs showed any bias in their sub-sampling distribution, i.e., any significant difference between the original sample and the reduced sample of 35,527 participants who had values for all 18 variables considered (the 12 best MRFs and 6 confounders: age, sex, age$^2$, age × sex, age$^2$ × sex, head size; Supplementary Information). In total, the 12 best MRFs explained 1.5% of the effect on the vulnerable brain network ($F_{12;35509} = 43.5$).

While 6 out of the 7 genetic clusters associated with the LIFO network were correlated with many variables related to each of the 15 MRF categories, including diabetes, alcohol consumption and traffic pollution (Supplementary Data 1), we also found some genetic overlap between the very specific best MRF of "alcohol intake frequency" and the LIFO network in the pleiotropic rs13107325 variant (cluster 2), as well as rs17690703, part of the large genetic cluster 5 in MAPT (Supplementary Data 4). No genetic overlap was found for the precise "nitrogen dioxide air pollution in 2005" or "diabetes diagnosed by doctor", nor for approximate variables.

## Discussion

This study reveals, in a cohort of nearly 40,000 UK Biobank participants, the genetic and modifiable risk factors' associations with brain regions in a 'last in, first out' (LIFO) network that show earlier and accelerated ageing and are particularly vulnerable to disease processes such as that of Alzheimer's disease[8]. Seven genetic clusters, two of which in the pseudo-autosomal region of the sex chromosomes coding for two antigens of the XG blood system, were found significantly associated and replicated genome-wide. In addition, after accounting for age and sex effects, diabetes, traffic-related pollution and alcohol were the most deleterious modifiable risk factors (MRFs) on these particularly vulnerable brain regions.

Three lead variants for our significant genetic clusters have been previously associated with ageing-related brain imaging measures in recent studies: one, in cluster 1, an eQTL of KCNK2 (TREK1)[12,13], whose increase in expression mediates neuroprotection during ischemia[14], the ubiquitous rs13107325 (cluster 2), and one, in cluster 4, in an intron of NUAK1 (ARK5)[15–17], which has been associated with tau pathology[18] (Table 1/Supplementary Data 1). On the other hand, of the seven genetic clusters, three were entirely novel (clusters 3, 6 and 7), and not found in other brain imaging studies, including our most recent work that expanded on our previous GWAS of all of the brain IDPs available in UK Biobank[19] by including more participants—in fact, the same number of participants as analysed in this present work—and, crucially, by also including the X chromosome[20] (Table 1/Supplementary Data 1). This suggests that, beyond the genetic hits that were meaningfully associated with the LIFO brain network and an array of relevant risk factors, lifestyle variables and brain disorders, and found in a few other imaging GWAS, some of the genetic underpinnings of the LIFO network are intrinsically specific to it and to no other pre-existing imaging phenotype.

All five autosomal genetic clusters identified through the GWAS of the LIFO phenotype had relevant associations with risk factors for dementia (Results; Supplementary Data 1), including precisely two of the best MRFs (for clusters 2 and 5), and three of them directly related in UK Biobank to the two diseases showing a pattern of brain abnormalities following the LIFO network: schizophrenia (clusters 2 and 4) and Alzheimer's disease (cluster 1) (Supplementary Data 1). In

particular, cluster 2 has its lead variant rs13107325 in an exon of one of the most pleiotropic genes ZIP8, which codes for a zinc and metal transporter. Considering the vulnerability of the LIFO brain network to adolescent-onset schizophrenia and its significant association with fluid intelligence that we previously demonstrated[8], it is notable that this variant has been associated genome-wide with schizophrenia[6], as well as intelligence, educational attainment and mathematics ability[5,21]. In line with the LIFO brain network being both prone to accelerated ageing and susceptible to Alzheimer's disease, this genetic locus has also been associated genome-wide with well-known risk factors for dementia. These comprise alcohol—including the exact same variable of "alcohol intake frequency" as identified as one of the best MRFs—cholesterol, weight, sleep—including "sleep duration"—and blood pressure[22–26], all of which significantly contribute to modulating the LIFO brain network when considered separately (Table 2/Supplementary Data 3). Of relevance, this genetic locus is also associated to an increased risk of cardiovascular death[27]. Cluster 5, a large genetic cluster in the MAPT region (Microtubule-Associated Protein Tau), comprised in total 3906 significant variants (Supplementary Data 4). This genetic region plays a role in various neurodegenerative disorders related to mutations of the protein tau, such as frontotemporal dementia[28] and progressive supranuclear palsy[29], but also, of particular pertinence to the LIFO brain network, Alzheimer's and Parkinson's disease, with a genetic overlap between these two diseases in a locus included in our significant cluster 5 (rs393152, $\beta = -0.09$, $P = 6.35 \times 10^{-14}$)[4]. Despite the relatively low number of people with diagnosed Alzheimer's disease in the genetic discovery cohort, we were able to establish—albeit with small effect sizes—a significant mediation role for the LIFO brain regions between the lead bi-allelic variant for cluster 5 and this Alzheimer's diagnosis, suggesting once more the importance played by these vulnerable brain areas in unhealthy ageing.

Finally, of the seven clusters, two were located in the pseudo-autosomal region (PAR1) of the sex chromosomes corresponding to the genes XG and CD99, coding for the two antigens of the XG blood group. This blood group system has been largely neglected, its main contribution related to the mapping of the X chromosome itself, and its clinical role remains elusive[30]. In order to investigate further the possible role of these two variants of the XG blood group, we examined out-of-sample their associations with thousands of non-imaging phenotypes. This analysis revealed that the first of these two loci was significantly and consistently associated with early life factors, environmental factors and health outcomes, including particulate matter and nitrogen dioxide air pollution, the second most deleterious MRF to the LIFO brain network (Supplementary Data 2). Whether these associations are due to stratification or genotyping artefacts, or to the fact that this specific variant, which is inherited from a parent, has a parental impact that modulates the effect of early life environment of the UK Biobank participants, the so-called "nature of nurture", will need further investigation[31].

Intriguingly, an analysis revealed that the genes involved in the loci associated with the LIFO network (Table 1/Supplementary Data 1) are enriched for the gene ontology terms of leucocyte extravasation, namely "positive regulation of neutrophil extravasation" ($P = 4.75 \times 10^{-6}$) and "T cell extravasation" ($P = 4.75 \times 10^{-6}$). This result held when removing the genes included in the MAPT extended region (with $P = 2.54 \times 10^{-6}$ and $P = 2.54 \times 10^{-6}$, respectively). Leucocyte extravasation facilitates the immune and inflammatory response, and there has been renewed focus on the fact that a breakdown of the blood-brain barrier together with leukocyte extravasation might contribute to both Alzheimer's disease and schizophrenia[32,33]. In line with the enrichment findings, 4 out of the 7 genetic clusters associated with the LIFO network are correlated in UK Biobank blood assays with percentage or count of immune cells (neutrophil, lymphocyte, platelet, monocyte, etc.; Supplementary Data 1).

Regarding MRFs' effects on the LIFO brain network, diabetes and alcohol consumption have been consistently shown to be associated with both cerebral and cognitive decline[34,35]. On the other hand, pollution—and notably that of nitrogen oxides—has emerged more recently as a potential MRF for dementia[2,36]. In particular, the increase of dementia risk due to nitrogen oxide pollution, a proxy for traffic-related air pollution, seems to be enhanced by cardiovascular disease[37]. In this study, we found that nitrogen dioxide pollution has one of the most deleterious effects onto the fragile LIFO brain regions. This effect could only be unmasked by regressing out the effects of age and sex, as traffic-related air pollution is modestly inversely-correlated with age (Supplementary Data 5). It is also worth noting that including age and sex as confounding variables in the first stage of our analysis reduced considerably the contribution of what had appeared at first— before regression—as the most harmful risk factors: blood pressure, cholesterol and weight (Table 2/Supplementary Data 3). Furthermore, the benefit of examining these MRFs in a single model in the second stage of our analysis is that we can assess the unique contribution of each of these factors on the LIFO brain network; in doing so, blood pressure, cholesterol and weight were no longer significant (Table 3).

One defining characteristic of the LIFO brain network is how much age explains its variance. Indeed, in the dataset covering most of the lifespan that was initially used to identify the LIFO and spatially define it[8], age explained 50%. In the UK Biobank imaging project, where imaged participants are over 45 years old, age explained 30% (Fig. 1). It is thus perhaps unsurprising that, while the explained variance by each of the MRFs varies widely (Table 2/Supplementary Data 3), it reduces notably once the effect of age and other confounders has been regressed out (without confounders included in the model: maximum 8.4%; with confounders: maximum 0.5%). Combined, the 12 best MRFs explained a significant 1.5% of the effect on the vulnerable brain network after regressing out age, head size and sex effects. Regarding the genetic hits, we found a significant heritability with $h^2 = 0.15$, in keeping with our results for structural brain phenotypes (except for subcortical and global brain volumes, which demonstrate higher heritability[19]).

The uniqueness of this study relies on the fact that we combined the strengths of two different cohorts: the first, which revealed the LIFO grey matter network, is lifespan, demonstrating the mirroring of developmental and ageing processes in the LIFO brain areas, something that could never be achieved with UK Biobank because of its limited age range. Of note, for this initial work with the lifespan cohort[8], we not only included grey matter partial volume images, as done in this current study, but also Freesurfer information of cortical thickness and surface area. The LIFO network showed no contribution from Freesurfer cortical thickness or area. This might hint at processes that only partial volume maps are able to detect due to the LIFO network's specific localisation, including in the cerebellum and subcortical structures, which are not included in the area and thickness surface methods from Freesurfer.

Limitations of our study pertain to the nature of the data itself and the way each variable is encoded in the UK Biobank (binary, ordinal, categorical, continuous), the number of missing values, what is offered as variables for each modifiable risk factor category (e.g. we chose not to create any compound variables, such as the ratio of cholesterol levels or systolic and diastolic blood pressures), and the curation of each of these variables. Some of the factors might be proxies for another category, but including the 'best' ones in a single model alleviate these issues to some extent. Another limitation is the assumption in our models that each risk factor has a linear, additive effect on the vulnerable LIFO brain network. It is also important to note that cross-sectional and longitudinal patterns of brain ageing can differ, as has been shown for instance for adult span trajectories of episodic and semantic memory, especially in younger adults[38]. A recent study has also demonstrated a specific 'brain age' imaging measure to be more related to early life influences on brain structure than within-person

rates of change in the ageing brain[39]. Further work will be needed to establish how the LIFO network data changes in terms of within-person trends, for instance by investigating the growing UK Biobank longitudinal imaging database. While we took care of assessing the replicability of our genetic results by randomly assigning a third of our dataset for such purposes (all our significant genetic hits were replicated), this was performed within the UK Biobank cohort that exhibits well-documented biases, being well-educated, less deprived, and healthier than the general population, especially for its imaging arm[40]. Independent replications will be needed to confirm the existence of the LIFO-associated genetic loci.

In conclusion, our study reveals the modifiable and nonmodifiable factors associated with some of the most fragile parts of the brain particularly vulnerable to ageing and disease process. It shows that, above and beyond the effect of age and sex, the most deleterious modifiable risk factors to this brain network of higher-order regions are diabetes, pollution and alcohol intake. Genetic factors are related to immune and inflammatory response, tau pathology, metal transport and vascular dysfunction, as well as to the XG blood group system from the pseudo-autosomal region of the sex chromosomes, and meaningfully associated with relevant modifiable risk factors for dementia. The unprecedented genome-wide discovery of the two variants on the sex chromosomes in this relatively unexplored blood group opens the way for further investigation into its possible role in underlying unhealthy ageing.

Supplementary Information is available for this paper.

## Methods

For the present work the imaging cohort of UK Biobank was used and we included 39,676 subjects who had been scanned and for whom the brain scans had been preprocessed at the time of the final set of analyses (M/F 47–53%; 44–82 years, mean age 64 ± 7 years; as of October 2020)[41,42]. Structural T1-weighted scans for each participant were processed using the FSL-VBM automated tool to extract their grey matter map[43,44]. The 'last in, first out' (LIFO) network of mainly higher-order brain regions was initially identified by performing a linked independent component analysis on the grey matter images of another, lifespan observational cohort of 484 subjects[8,45,46]. This map of interest, along with the other 69 generated by the analysis, was first realigned to the UK Biobank 'standard' space defined by the grey matter average across the first 15,000 participants, then regressed into the UK Biobank participants' grey matter data, to extract weighted average values of grey matter normalised volume inside each of the z-maps, using the z-score as weighting factor. This made it possible to assess the unique contribution of this specific LIFO map, above and beyond all the rest of the brain represented in the other 69 maps. At the end of this process, we obtained a single imaging measure for each of the 39,676 participants, i.e. a 'loading' corresponding to their amount of grey matter normalised volume in the LIFO brain network.

### Ethics

Human participants: UK Biobank has approval from the North West Multi-Centre Research Ethics Committee (MREC) to obtain and disseminate data and samples from the participants (http://www.ukbiobank.ac.uk/ethics/), and these ethical regulations cover the work in this study. Written informed consent was obtained from all of the participants.

### Modifiable risk factors selection

The following 15 categories of modifiable risk factors (MRFs) for dementia were investigated based on previous literature: blood pressure, diabetes, cholesterol, weight, alcohol, smoking, depression, hearing, inflammation, pollution, sleep, exercise, diet/supplementation, socialisation, and education. These included well-documented cerebrovascular risk factors, and in particular included all of the 12

modifiable risk factors considered in the updated Lancet commission on dementia, with the sole exception of traumatic brain injury[3]. For each category, several MRF variables from UK Biobank were very minimally pre-processed (Supplementary Information). In total, 161 MRF variables were obtained. To optimise the interpretability of the results, and to be able to relate them to previous findings, we did not carry out any data reduction, which would have prevented us from identifying exactly which variable—and subsequently, which genetic component for this specific variable—contribute to the effect. For these same reasons, we did not create any compound variable.

## Statistical analyses

**Genome-wide association study.** We followed the same protocol we had developed for the first genome-wide association study (GWAS) with imaging carried out on UK Biobank[19]. Briefly, we examined imputed UK Biobank genotype data[47], and restricted the analysis to samples that were unrelated (thereby setting aside only ~450 participants), without aneuploidy and with recent UK ancestry. To account for population stratification, 40 genetic principal components were used in the genetic association tests as is recommended for UK Biobank genetic studies[19,20,47]. We excluded genetic variants with minor allele frequency <0.01 or INFO score <0.03 or Hardy-Weinberg equilibrium $-\log_{10}(P) > 7$. We then randomly split the samples into a discovery set with 2/3 of the samples ($n = 22,128$) and a replication set with 1/3 of the samples ($n = 11,083$). We also examined the X chromosome with the same filters, additionally excluding participants with sex chromosome aneuploidy: 12 in non-pseudoautosomal region (PAR) and 9 in PAR for the discovery set, 3 in non-PAR and 6 in PAR for the replication set. Variants were considered significant at $-\log_{10}(P) > 7.5$, and replicated at $P < 0.05$.

**Modifiable risk factor study.** In the first stage, the general linear model was used to investigate, separately, the association between each of these 161 MRFs and the LIFO network loadings in all the scanned UK Biobank participants ($n = 39,676$). We ran each model twice: once as is, and once adding 6 confounders: age, $age^2$, sex, age × sex, $age^2$ × sex, and head size, to estimate the contribution of these MRFs on the LIFO network above and beyond the dominating effects of age and sex. Sex was based on the population characteristics entry of UK Biobank. This is a mixture of the sex the NHS had recorded for the participant at recruitment, and updated self-reported sex. For the GWAS, both sex and genetic sex were used (the sample was excluded in case of a mismatch). In total, 32 variables tailored to structural imaging had been considered as possible confounders, and we retained those with the strongest association ($R^2 \geq 0.01$; see Supplementary Information). Socioeconomic status via the Townsend deprivation index was also considered as a possible confounding variable but explained little variance ($R^2 < 0.001$) and thus was not included as a confounder.

MRFs were not considered further if they were not significant—not surviving Bonferroni-correction, i.e., $P > 1.55 \times 10^{-4}$—and if more than 5% of the subjects had their MRF values missing. For each category, a single 'best' MRF was then selected as the variable with the highest $R^2$ among those remaining, after regressing out the confounding effects of age and sex.

In the second stage, all these best MRFs were then included in a single general linear model, together with the same 6 confounders used in the first stage, to assess the unique contribution of each factor on the LIFO brain network loadings. A prerequisite to carry out this single general linear model analysis was to only include participants who would have values for all best MRFs and confounders. This explains the additional criterion of only including MRFs that had no more than 5% of values missing, to ensure that the final sample of participants who had values for all these best and confounding factors would not be biased compared with the original sample—something

we formally tested (see Supplementary Information)—especially as data are not missing at random in UK Biobank, and exhibit some genetic structure[48]. The sample was therefore reduced to a total of 35,527 participants for this second stage analysis (M/F 17,290–18,237; 45–82 years, mean 64 ± 7 years). The effect of these best MRFs taken altogether was considered significant with a very conservative Bonferroni correction for multiple comparisons across all combinations of every possible MRF from each of the initial 15 MRF categories ($P < 4.62 \times 10^{-17}$, see Supplementary Information for more details). In addition, both full and partial correlations were computed for the same set of best MRFs and confounders, in order to assess possible relationships between variables.

## Post hoc genetic analyses

**Chromosome 17 inversion.** We investigated chromosome 17 inversion status of the participants in the discovery cohort by considering their genotype on 32 variants that tag chromosome 17 inversion according to Steinberg et al.[11]. Of these 32 variants, 24 were present in our genetic data. We labelled the participants homozygous inverted, heterozygous, or homozygous direct (not inverted) when all 24 of these alleles indicated the same zygosity. This yielded an unambiguous inversion status for 21,969 participants (99% of the discovery cohort). To examine if the association between the non-triallelic lead variant of the *MAPT* genetic cluster (rs2532395, Table 1/Supplementary Data 1) and the LIFO network was independent from this common inversion, we determined inversion/direct status of the discovery cohort and: 1. repeated the association test between rs2532395 and the LIFO phenotype, with chromosome 17 inversion status added as a confounder; and 2. correlated the genotype for rs2532395 with chromosome 17 inversion.

**Causality within each genetic cluster.** We used CAVIAR (Causal Variants Identification in Associated Regions[49]) to assess causality of variants that passed the genome-wide significance threshold in each of the genetic clusters we report. CAVIAR uses a Bayesian model and the local linkage disequilibrium structure to assign posterior probabilities of causality to each variant in a region, given summary statistics for an association. We did not perform CAVIAR analysis on the genetic cluster on chromosome 17, as its non-triallelic lead variant (rs2532395) was strongly correlated with chromosome 17 inversion, and the LD matrix was large and low rank. We excluded the X chromosome loci from this analysis due to the difficulty in assessing LD in this chromosome.

**Enrichment analysis.** Based on the genes listed in the 'Genes' column of Table 1/Supplementary Data 1, we performed an enrichment analysis for the genes associated with the LIFO brain network using PANTHER[50]. PANTHER determines whether a gene function is overrepresented in a set of genes, according to the gene ontology consortium[51,52].

**Mediation analysis between MAPT top variant and Alzheimer's disease, via the LIFO brain network.** As the gene *MAPT* is associated with Alzheimer's disease, and as we found a significant association between *MAPT* and the LIFO brain network, we examined to what extent the effect of *MAPT* is mediated by the LIFO brain regions. We conducted a mediation analysis using the counterfactual framework in which the average indirect effect of the treatment on the outcome through the mediator is nonparametrically identified (version 4.5.0 of the R package 'mediation'[53]). This is a general approach that encompasses the classical linear structural equation modelling framework for causal mediation, allowing both linear and non-linear relationships. In this analysis, the genotype for the lead bi-allelic variant of the *MAPT* association was used as the treatment, the LIFO loadings as the mediator, and Alzheimer's disease diagnosis as the outcome.

From the ~43 K UK Biobank participants who had been scanned, we searched for those who had been diagnosed with Alzheimer's

disease specifically, regardless of whether this diagnosis occurred before, or after their brain scans. Based on hospital inpatient records (ICD10: F000, F001, F002, F009, G300, G301, G308, and G309 and ICD9: 3310) and primary care (GP) data (Eu00., Eu000, Eu001, Eu002, Eu00z, F110., F1100, F1101, Fyu30, X002x, X002y, X002z, X0030, X0031, X0032, X0033, XaIKB, XaIKC, and XE17j), we identified 65 such cases− UK Biobank being healthier than the general population, and those scanned showing an even stronger healthy bias−of which 34 were included in the discovery set after QC.

We considered two conditions for the effect of the treatment on the outcome. First, a dominant condition in which the minor allele is assumed to be dominant and for which at least one copy of the minor allele is considered treated. Second, a recessive condition in which the minor allele is assumed to be recessive. We considered that either condition was nominally significant if the confidence interval of the average causal mediated effect did not intersect zero, and had an associated $P < 0.05 \div 2$ (correcting for the two conditions). We assessed confidence intervals and $P$-values using 50,000 bootstrapped samples.

**Associations between the LIFO brain network's genetic hits and the MRFs.** First, we reported in Table 1/Supplementary Data 1 the significant associations between the LIFO genetic hits and UK Biobank variables related to the 15 categories listed for the MRFs. For this, we used the Open Targets Genetics website, which reports the GWAS carried out in UK Biobank (https://genetics.opentargets.org/). Second, we assessed whether there was any genetic overlap between the known genetic components of the 3 best MRFs and the LIFO phenotype. Again, we used the Open Targets Genetics website outputs for these 3 very specific UK Biobank variables, and compared the significant hits for these 3 best MRFs within ±250 kbp of, or in high LD (>0.8) with, our own LIFO variants. If reported hits were limited, we also searched online for GWAS done on similar variables. Finally, we also included the list of significant hits for diabetes[54], which focused on a potential genetic overlap between diabetes and Alzheimer's disease.

**Post hoc association for the sex chromosomes variants.** The allele counts of each participant for two specific significant variants of the sex chromosomes not−or hardly−available in open databases such as https://genetics.opentargets.org/[55] were further associated out-of-sample with all non-imaging phenotypes of UK Biobank ($n = 16,924$). This analysis was carried out in the entire genotyped, quality-controlled sample where participants who had been scanned were removed (final sample: 374,230 participants), taking into account the population structure (40 genetic principal components), as well as the confounding effects of age, sex, age x sex, age$^2$ and age$^2$ x sex. Results were corrected for multiple comparisons across all non-imaging phenotypes and the two variants.

**Heritability.** We examined the heritability of the LIFO phenotype, and the coheritability between the LIFO network and Alzheimer's disease or schizophrenia using LDSC[56]. This method uses regression on summary statistics to determine narrow sense heritability $h^2$ of a trait, or the shared genetic architecture between two traits. LDSC corrects for bias LD structure using LD calculated from a reference panel (we used LD from the Thousand Genomes Project Phase 1[57]). We obtained summary statistics for a meta-analysis of Alzheimer's disease involving 71,880 cases and 383,378 controls[58]. The number of genetic variants in the intersection between the summary statistics was 1,122,435. For schizophrenia, the summary statistics were obtained from a meta-analysis involving 53,386 cases and 77,258 controls[59]. A total of 1,171,319 genetic variants were in the intersection with the summary statistics for LIFO. For both Alzheimer's and schizophrenia, the X chromosome was not included in the heritability calculation, as it was

excluded from the meta-analysis that we sourced the summary statistics from.

**Reproducibility.** No data was excluded for the MRF analyses. For the genetic analyses, these were restricted to samples that were unrelated, without aneuploidy and with recent UK ancestry (see above).

No statistical method was used to predetermine sample size. The experiments were not randomised. The Investigators were not blinded to allocation during experiments and outcome assessment.

**Reporting summary**
Further information on research design is available in the Nature Portfolio Reporting Summary linked to this article.

## Data availability
All the FLICA decomposition maps − including the LIFO grey matter network − in UK Biobank standard space, the UK Biobank grey matter template, scripts, and the LIFO loadings for all of the participants are freely available on a dedicated webpage: open.win.ox.ac.uk/pages/douaud/ukb-lifo-flica/.

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

## Acknowledgements

We are grateful to Profs Christian K. Tamnes, Lars T. Westlye, Kristine B. Walhovd and Anders M. Fjell, and Dr Andreas Engvig for providing the lifespan cohort which was used to initially derive the original 'last in, first out' brain network map, and to Prof Augustine Kong for helpful discussion on the associations between the PAR hit and early life and environmental factors. G.D. was supported by a UK MRC Career Development Fellowship (MR/K006673/1) and a Wellcome Collaborative Award (215573/Z/19/Z). S.S. was supported by Wellcome (203139/Z/16/Z; 215573/Z/19/Z). L.E. was funded by NSERC grants (RGPIN/05484-2019; DGECR/00118-2019) and a Michael Smith Health Research BC Scholar Award. A.M.W. received support through the NIH Intramural Research Program (ZIA-MH002781; ZIA-MH002782). This research was funded in whole, or in part, by the Wellcome Trust (215573/Z/19/Z; 203139/Z/16/Z; 203139/A/16/Z). For the purpose of Open Access, the author has applied a CC BY public copyright licence to any Author Accepted Manuscript version arising from this submission. This research was also supported by the NIHR Oxford Health Biomedical Research Centre (NIHR203316). The views expressed are those of the author(s) and not necessarily those of the NIHR or the Department of Health and Social Care. The Wellcome Centre for Integrative Neuroimaging is supported by core funding from the Wellcome Trust (203139/Z/16/Z and 203139/A/16/Z).

## Author contributions

G.D. conceived and supervised the work, and carried out some of the genetic and modifiable risk factors analyses. J.Ma. carried out most of the genetic and modifiable risk factors analyses. J.Mi., S.L., A.M.W., and L.T.E. carried out additional genetics analyses. G.D., P. McC., F.A.-A., S.S., and L.T.E. created/extracted the imaging and genetics data, and organised the non-imaging data and confound variables. L.T.E. co-supervised the genetic analyses. A.M.W. co-supervised the modifiable risk factor analyses. G.D. interpreted the results and wrote the paper. J.Ma., S.S., L.T.E., and A.M.W. revised the paper.

## Competing interests

The authors declare no competing interests.

## Additional information

[1]FMRIB Centre, Wellcome Centre for Integrative Neuroimaging (WIN), Nuffield Department of Clinical Neurosciences, University of Oxford, Oxford, UK. [2]FOCUS Lab, Department of Psychology, University of Turin, Turin, Italy. [3]Department of Statistics and Actuarial Science, Simon Fraser University, Burnaby, Canada. [4]Pacific Parkinson's Research Centre, The University of British Columbia, Vancouver, BC, Canada. [5]National Institutes of Mental Health, National Institutes of Health, Bethesda, MD, USA. [6]Department of Human Genetics, University of Texas Rio Grande Valley, Brownsville, TX, USA. [7]These authors contributed equally: Lloyd T. Elliott, Anderson M. Winkler. ✉e-mail: gwenaelle.douaud@ndcn.ox.ac.uk

