## [Peer Review File · Nature Communications]

The effects of genetic and modifiable risk factors on brain regions vulnerable to ageing and diseaseREVIEWER COMMENTS

Reviewer #1 (Remarks to the Author):

Manuello et alia presents a nice re-analysis of UKbiobank MRI dataset to assess genetic and modifiable risk factor influencing the LIFO vulnerable phenotype they previously described. The manuscript is original and interesting although generalization and validation of their findings is certainly unknown. I feel that the authors should look for the way to identify independent datasets for corroborating their current findings in the magnificent UKbiobank dataset. Otherwise, their findings will remain as observations that need a further corroboration.

Main issue with any new GWAS hits is that once GWAS significance is reached, the field request at least two independent replications for declaring proven the hits. So, LIFO associated genetic loci described herewith will remain as "pending" until other cohort is appropriately scrutinized with identical methodology and the same hits with a comparable effect size a direction appear again. This is a limitation that the authors must declare in their manuscript. The hits described are unconfirmed.

Another important point to be addressed is the variance of LIFO explained by MRF and genetic hits separately and together. This is, what is the amount of variance explained by common variants (GWAS heritability), MRFs, age etc? This is an important point to be disclosed, explored and widely discussed in currently the text. If the variance of LIFO phenotype is mostly unmodelled with currently available variables, please explain the potential reasons of it and try also to explore the potential (inherent) stochasticity of LIFO phenotype itself.

Regarding the MAPT locus, the relationships of their findings with the H1/H2 haplotypes tracking the ancestral inversion observed in this chromosomal region should be addressed (<https://www.nature.com/articles/ng1508>). Are the LIFO hits independent to the common chr 17 inversions observed in humans? Colocalization with neurodegenerative loci previously described in this region is also important for the field. Mediation analysis of the LIFO phenotype for the effect of MAPT locus in proxy-AD can be also explored in the UKbiobank dataset.

Finally, the authors are in an optimal position to address if the genetic components of top MRF detected for LIFO are also associated with the LIFO phenotype. This is especially relevant for diabetes because an enormous controversy exists about the genetic connection between Alzheimer/Neurodegenerative diseases and diabetes risk susceptibility loci (as an example see a very recent report by Escott Price et al. [https://doi.org/10.1016/s1474-4422\(22\)00395-7](https://doi.org/10.1016/s1474-4422(22)00395-7) and references therein).

Reviewer #2 (Remarks to the Author):

Re: Genetic and modifiable risk factors' effects on brain regions most vulnerable to ageing and disease (ref NCOMMS-23-06958)

This manuscript presents an interesting research question of clinical relevance: the identification of common alleles and 'modifiable' factors influencing the neuroanatomy of a group of brain regions shown to have an age-related slower development, but a quicker degeneration (named here 'Late In, First Out' [LIFO]). This group of brain regions have also shown spatial correlation with changes seen in participants with Alzheimer and Schizophrenia; suggesting being a 'vulnerable' set of regions for neurodevelopmental/neurodegenerative disorders. Consequently, the authors run to separate analyses: A GWAS of averaged volume of the brain regions within this network, and a series of GLM analyses to examine the association between different 'modifiable' factors and the same brain phenotype. Despite the methods applied are sound, I feel that the analyses conducted do not fully address the question posed and they only scratch the surface of the study's aim. As such, I have doubts about the real significance and impact of the results reported.

I understand that the authors base their approach on a previous paper where they identified the LIFO network, and that this was done based on regional brain volumes. Consequently, they calculate a weighted average grey matter volume measure for the LIFO. However, from a genetic

perspective, cortical volume is a less interesting measure – at least to me – than their components of cortical thickness and surface area. We know that these two cortical metrics are under different genetic influences, showing little genetic correlation (Grasby et al, Science 2020). It would be interesting to realise whether cortical thickness and/or surface area in this brain network present with quadratic developmental trend, in other words, if any of the two is a major contributor to this network characteristic trend. Then, a GWAS of the selected metric, or two separate GWASes for each of them if both show to contribute, would have increased interest to me. Moreover, I would be interested in knowing how the results compare to those of Grasby et al., who conducted GWAS not only for overall thickness and surface area, but also for all cortical gyri based on the Desikan-Killiany atlas separately. Furthermore, postGWAS basic analyses (e.g. finemapping) would add interest to the results, as well as the investigation of genetic correlations with other phenotypes (particularly Alzheimer's disease and schizophrenia). In my opinion, this section of the manuscript, taken to a far deeper investigation than currently presented, would represent in itself a very interesting contribution to the field.

Instead of deepening their GWAS analysis, the authors present a separated analysis on the association between 'modifiable' risk factors and LIFO's weighted average grey matter volume. I do have reservations about the interest of this analysis and how it is conducted. It does not appear clear to me how the 15 categories in which variables are grouped are defined, neither how each variable is assigned to them? Why did the authors not enter in their regression all variables with missingness above their selected criteria? Only taking one variable per category based on the lowest p value does not appear the best criteria to me, considering that all association values are very small and in most cases with neglectable differences among them, and since these are models with a single predictor of interest, the significant variables discarded might have been explaining separate variance from the one selected? Some of these 'modifiable' factors are under a significant genetic influence and might have been simply more interesting to determine to what extent there's genetic overlap between common alleles influencing these and the GWAS results for LIFO's weighted average volume? ... I do not really understand this second part of the manuscript and how it all fits with the GWAS analysis done previously.

Another aspect that surprised me of this manuscript was the fact that only sex, age and brain volume were added as covariates in the analyses. However, the authors have demonstrated the importance of accounting for many other confounders in UK Biobank's imaging data such as head movement, head position, table position, assessment centre, etc. (Alfaro-Almagro et al, 2021 Neuroimage).

Reviewer #3 (Remarks to the Author):

Review of NCOMMS-23-06958

The present manuscript reports the genetic and phenotypic correlates of the previously-characterised LIFO brain network (grey matter loci whose grey matter is most strongly correlated with cross-sectional age differences using a VBM-based ICA) in the UK Biobank imaging sample. Associations with this network across the genome (GWAS) and across a large number of raw modifiable risk factors (MRFs) are reported, with relatively few, but plausible, hits of small effect size across both sets of results. I like very much that the LIFO network as a quantitative response to the retrogenesis hypothesis/observations, and it is one valuable and important approach for distilling out those structural properties which are most interesting to subject to the types of testing presented here. These are reasonable questions to ask, and the manuscript has several strengths including LIFO having been trained out of sample, the desire to seek GWAS replication (within two UKB subsets), and an interest in multivariable analyses to identify unique contributions of MRFs, though there are a number of analytical and interpretative issues which I raise below.

- Sincere apologies if I have missed it, but I could not see that the authors had taken steps to control for technical covariates in their genetic analyses. Typically batch, array and PCs for population stratification.

- GWAS analyses – can the authors provide further information on why they opted to discard data/power on related individuals when it is possible to preserve power and model kinship?
- There doesn't seem to be any acknowledgement that cross-sectional data were used as a proxy for longitudinal ageing effects (both in training and testing). Is it known how well the LIFO network data can be reconciled with within-person trends?
- The LIFO was trained on a smaller cross-sectional dataset some time ago. I recognise the value in having lifecourse developmental MRI data, not available in UKB. However, with the much larger UKB sample in mind, to what extent are the near-absent outer-cortical loci in the LIFO mask due to gyral folding/statistical power differences across the brain space in the initial discovery cohort? Would it boost power/precision for the GWAS and RF analyses to use a UKB discovery sample to re-run the VBM ICA? Whether it yields fundamentally different loci is an empirical question. In fact, are areas which are ageing-related but not important for development of additional interest (though of course, this later analyses might be beyond the scope of the present paper)?
- The authors report some genetic clusters whose associations with brain measures have been reported previously. Similarly, many of the associations between the top MRFs and brain measures have been identified previously both in UKB and other samples. Whereas it is good to have replicated those findings, to what extent is the LIFO weighted mask just telling us about overall grey matter ~ genotype/phenotype associations? That is, can the authors draw out more clearly (in empirical terms) for the reader that LIFO is getting at something statistically more informative – with respect to gene and RF - than one could identify examining all GM, or variants thereof (such as sulcal widening or brain age) which have previously reported some GWAS hits reported here?
- Modifiable RF study: Given several co-authors of the current study conducted a detailed study highlighting the effect of modelling imaging confounds in UK Biobank, the use of a very restricted set in the current analyses – just age and sex (++) and head size (it that skull scaling?) – could do with a brief additional comment.
- State briefly in the text whether the directions of associations between rs312238 and the mentioned phenotypes are in a consistent/expected direction.
- Suggest more careful phrasing of causal language throughout – e.g. “modifiable risk factors’ influences over the vulnerable LIFO brain network”
- The title seems to refer to the LIFO as being *the* brain network most vulnerable to disease. I am not sure that has been empirically demonstrated, has it? On a related note, does the absence of APOE associations here – a well-known genetic risk factor for brain and cognitive ageing, and dementia – potentially speak to the lack of statistical power for the GWAS setup (which may have been reduced by opting for a discovery/replication sample rather than a single-sample analysis)?

First, we would like to apologise for the delay in revising this manuscript due to personal circumstances. We are grateful to the reviewers for their positive remarks and constructive suggestions, which we have addressed below. Modifications have been highlighted in the manuscript in yellow.

In particular, we now: carry out an analysis of the role for the chromosome 17 inversion in our results; assess the heritability of our phenotype of interest (the LIFO brain network), as well as its shared heritability with Alzheimer's disease and schizophrenia; include a new mediation analysis between our *MAPT* result and Alzheimer's disease via the LIFO brain network; investigate the genetic overlap between the LIFO network and modifiable risk factors, and expand Table 1 accordingly; considerably increase the section on the link between genetics and risk factors; assess causality among the significant variants for each autosomal genetic cluster; and carry out an enrichment analysis of the genes involved with our significant findings. We strongly believe that these changes have significantly improved the paper, and in particular tightened the link between genetic and modifiable risk factors.

Reviewer #1:

Manuello et alia presents a nice re-analysis of UKbiobank MRI dataset to assess genetic and modifiable risk factor influencing the LIFO vulnerable phenotype they previously described. The manuscript is original and interesting although generalization and validation of their findings is certainly unknown. I feel that the authors should look for the way to identify independent datasets for corroborating their current findings in the magnificent UKbiobank dataset. Otherwise, their findings will remain as observations that need a further corroboration.

We first would like to thank Reviewer 1 for this positive summary of our study.

- 1. Main issue with any new GWAS hits is that once GWAs significance is reached, the field request at least two independent replications for declaring proven the hits. So, LIFO associated genetic loci described herewith will remain as "pending" until other cohort is appropriately scrutinized with identical methodology and the same hits with a comparable effect size a direction appear again. This is a limitation that the authors must declare in their manuscript. The hits described are unconfirmed.*

We entirely agree that replication is an important component for interpreting GWAS outcomes. To this effect, we have split the data at random in our study and set aside a third of the participants' brain imaging scans (n = 11,083), for replication. **All of the 7 significant hits replicated** (see last column of **Table 1**).

We understand however that our out-of-sample replication was performed *within* the UK Biobank, which demonstrate well-known biases (Fry *et al.*, 2017). We have thus **expanded the section of the Discussion** where we consider the various limitations of this study:

Discussion: While we took care of assessing the replicability of our genetic results by randomly assigning a third of our dataset for such purposes (all our significant genetic hits were replicated), this was performed within the UK Biobank cohort that exhibits well-documented biases, being well-educated, less deprived, and healthier than the general population, especially for its imaging arm⁴¹. Independent replications will be needed to confirm the existence of the LIFO-associated genetic loci.

References: 41. Fry, A. et al. Comparison of Sociodemographic and Health-Related Characteristics of UK Biobank Participants With Those of the General Population. *Am J Epidemiol* 186, 1026-1034, doi:10.1093/aje/kwx246 (2017).

2. *Another important point to be addressed is the variance of LIFO explained by MRF and genetic hits separately and together. This is, what is the amount of variance explained by common variants (GWAS heritability), MRFs, age etc? This is an important point to be disclosed, explored and widely discussed in currently the text. If the variance of LIFO phenotype is mostly unmodelled with currently available variables, please explain the potential reasons of it and try also to explore the potential (inherent) stochasticity of LIFO phenotype itself.*

We appreciate the requests from Reviewer 1 for exploring the variance explained of the LIFO phenotype by e.g. the genetic hits, and by the modifiable risk factors (MRFs). One key feature of the LIFO brain network is precisely how much it is explained by age: in the original dataset covering most of the lifespan, age explained 50%; in the UK Biobank where imaged participants are over 45 years old, **age explained 30%** ($P < 2.23 \times 10^{-308}$; (first paragraph of the Results, and **Figure 1**). The explained variance by the MRFs thus varies widely, especially and unsurprisingly once the effect of age (and confounders) has been regressed out, something that is detailed in **Table 2**. Regarding the genetic hits, we have now performed a heritability analysis, and found a significant heritability with **$h^2 = 0.15$** , which is in keeping with our previous results for structural brain phenotypes (Elliott *et al.*, Nature 2018). We have **added this information in the Methods:**

Methods: Heritability. We examined the heritability of the LIFO phenotype, and the coheritability between the LIFO network and Alzheimer's disease or schizophrenia using LDSC⁵⁷. This method uses regression on summary statistics to determine narrow sense heritability h^2 of a trait, or the shared genetic architecture between two traits. LDSC corrects for bias LD structure using LD calculated from a reference panel (we used LD from the Thousand Genomes Project Phase 1⁵⁸).

And in the Results:

Results: We found that the heritability of the LIFO network was significant, with $h^2 = 0.15$ ($se = 0.01$).

And we **discuss the amount of variance explained** by both genetics and MRFs:

Discussion: One defining characteristic of the LIFO brain network is how much age explains its variance. Indeed, in the dataset covering most of the lifespan that was initially used to identify the LIFO and spatially define it⁸, age explained 50%. In the UK Biobank imaging project, where imaged participants are over 45 years old, age explained 30% (**Figure 1**). It is thus perhaps unsurprising that, while the explained variance by each of the MRFs varies widely (**Table 2**), it reduces notably once the effect of age and other confounders has been regressed out (without confounders included in the model: maximum 8.4%; with confounders: maximum 0.5%). Combined, the 12 best MRFs explained a significant 1.5% of the effect on the vulnerable brain network after regressing out age, head size and sex effects. Regarding the genetic hits, we found a significant heritability with $h^2 = 0.15$, in keeping with our results for structural brain phenotypes (except for subcortical and global brain volumes, which demonstrate higher heritability²⁰).

References: 8. Douaud, G. et al. A common brain network links development, aging, and vulnerability to disease. *Proceedings of the National Academy of Sciences of the United States of America* 111, 17648-17653, doi:10.1073/pnas.1410378111 (2014).

20. Elliott, L. T. et al. Genome-wide association studies of brain imaging phenotypes in UK Biobank. *Nature* 562, 210-216, doi:10.1038/s41586-018-0571-7 (2018).

57. Gazal, S., Marquez-Luna, C., Finucane, H. K. & Price, A. L. Reconciling S-LDSC and LDAK functional enrichment estimates. *Nat Genet* 51, 1202-1204, doi:10.1038/s41588-019-0464-1 (2019).

58. Genomes Project, C. et al. A global reference for human genetic variation. *Nature* 526, 68-74, doi:10.1038/nature15393 (2015).

3. *Regarding the MAPT locus, the relationships of their findings with the H1/H2 haplotypes tracking the ancestral inversion observed in this chromosomal region should be addressed (<https://www.nature.com/articles/ng1508>). Are the LIFO hits independent to the common chr 17 inversions observed in humans?*

We thank Reviewer 1 for pointing at this additional step to interpret our results in chromosome 17. We have now assessed chromosome 17 inversion by examining the H1/H2 markers listed in Steinberg *et al.*, 2012. We found that the genotype at our associated *MAPT* locus (rs2532395, the highest bi-allelic hit) was **strongly correlated with chromosome 17 inversion** (Pearson correlation $r = 0.98$, $P < 2 \times 10^{-16}$). When chromosome 17 inversion was included as a confounder in the association test for $LIFO \sim MAPT$, the strength of the association was much reduced. Based on this result, we have **added the following text to the Methods section:**

Methods: We investigated chromosome 17 inversion status of the participants in the discovery cohort by considering their genotype on 32 variants that tag chromosome 17 inversion according to

Steinberg *et al.*¹². Of these 32 variants, 24 were present in our genetic data. We labelled the participants homozygous inverted, heterozygous, or homozygous direct (not inverted) when all 24 of these alleles indicated the same zygosity. This yielded an unambiguous inversion status for 21,969 participants (99% of the discovery cohort). To examine if the association between the bi-allelic lead variant of the *MAPT* genetic cluster (rs2532395, **Table 1**) and LIFO is independent from this common inversion, we determined inversion/direct status of the discovery cohort and: 1. repeated the association test between rs2532395 and LIFO, with chromosome 17 inversion status added as a confounder; and 2. correlated the genotype for rs2532395 with chromosome 17 inversion.

And in the Results section:

Results: *MAPT* is in 17q21.31, a chromosomal band involved with a common chromosome 17 inversion¹². Adding chromosome 17 inversion status as a confounder reduced the significance of the association ($\beta = -0.15$, $P = 5.67 \times 10^{-2}$). Since the genotype for rs2532395 was also strongly correlated with chromosome 17 inversion in our dataset (Pearson correlation $r = 0.98$, $P < 2 \times 10^{-16}$), this would suggest that the association between *MAPT* and LIFO is not independent from chromosome 17 inversion.

References: 12. Steinberg, K. M. et al. Structural diversity and African origin of the 17q21.31 inversion polymorphism. *Nat Genet* 44, 872-880, doi:10.1038/ng.2335 (2012).

4 *Colocalization with neurodegenerative loci previously described in this region is also important for the field. Mediation analysis of the LIFO phenotype for the effect of MAPT locus in proxy-AD can be also explored in the UKbiobank dataset.*

This was indeed an intriguing possibility, and we have thus **pursued the mediation analysis** suggested here. From the ~43K UK Biobank participants who had been scanned, we searched for those who had been diagnosed with Alzheimer's disease specifically, regardless of whether this diagnosis occurred before, or after their brain scans ("proxy-AD"). We identified 65 such cases – the UK Biobank being healthier than the general population, and those scanned showing an even stronger healthy bias – of which 34 were included in the discovery set after QC. For those cases, we performed the mediation analysis using the top bi-allelic variant in the *MAPT* region (rs2532395). We **found a significant effect** for both recessive and dominant scenarios. We have now **added a section in the Methods** describing this new analysis:

Methods: *Mediation analysis between MAPT top variant and Alzheimer's disease, via the LIFO brain network.* As the gene *MAPT* is associated with Alzheimer's disease, and as we found a significant association between *MAPT* and the LIFO brain network, we examined to what extent the effect of *MAPT* is mediated by the LIFO brain regions. We conducted a mediation analysis using the counterfactual framework in which the average indirect effect of the treatment on the outcome through the mediator is nonparametrically identified (version 4.5.0 of the R package 'mediation'⁵³). This is a general approach that encompasses the classical linear structural equation modelling

framework for causal mediation⁵⁴, allowing both linear and non-linear relationships. In this analysis, the genotype for the lead bi-allelic variant of the *MAPT* association was used as the treatment, the LIFO loadings as the mediator, and Alzheimer's disease diagnosis as the outcome.

From the ~43K UK Biobank participants who had been scanned, we searched for those who had been diagnosed with Alzheimer's disease specifically, regardless of whether this diagnosis occurred before, or after their brain scans. Based on hospital inpatient records (ICD10: F000, F001, F002, F009, G300, G301, G308, and G309 and ICD9: 3310) and primary care (GP) data (Eu00., Eu000, Eu001, Eu002, Eu00z, F110., F1100, F1101, Fyu30, X002x, X002y, X002z, X0030, X0031, X0032, X0033, XaIKB, XaIKC, and XE17j), we identified 65 such cases – the UK Biobank being healthier than the general population, and those scanned showing an even stronger healthy bias – of which 34 were included in the discovery set after QC.

We considered two conditions for the effect of the treatment on the outcome. First, a dominant condition in which the minor allele is assumed to be dominant and for which at least one copy of the minor allele is considered treated. Second, a recessive condition in which the minor allele is assumed to be recessive. We considered that either condition was nominally significant if the confidence interval of the average causal mediated effect did not intersect zero, and had an associated $P < 0.05 \div 2$ (correcting for the two conditions). We assessed confidence intervals and P-values using 50,000 bootstrapped samples.

And in the Results:

Results: As this extended genetic region is known for its pathological association with many neurodegenerative disorders including Alzheimer's disease, we investigated whether the LIFO brain regions mediated the effect of the *MAPT* genetic cluster (using the lead bi-allelic variant rs2532395) on Alzheimer's disease (see **Methods**). Despite small average causal mediated effect (ACME) sizes, we found a significant effect for both the dominant model (ACME $\beta = 1.16 \times 10^{-4}$; 95% CI = $[5.19 \times 10^{-5}, 1.99 \times 10^{-4}]$; $P = 4 \times 10^{-5}$) and the recessive model (ACME $\beta = 1.55 \times 10^{-4}$; 95% CI = $[3.96 \times 10^{-5}, 3.74 \times 10^{-4}]$; $P = 4 \times 10^{-5}$; full output of the mediation package on the dominant and recessive models in **Table S1**).

As well as in the Discussion:

Discussion: Despite the relatively low number of people with diagnosed Alzheimer's disease in the genetic discovery cohort, we were able to establish – albeit with small effect sizes – a significant mediation role for the LIFO brain regions between the lead bi-allelic variant for cluster 5 and this Alzheimer's diagnosis, suggesting once more the importance played by these vulnerable brain areas in unhealthy ageing.

References: 54. Imai, K., Keele, L. & Tingley, D. A general approach to causal mediation analysis. *Psychol Methods* 15, 309-334, doi:10.1037/a0020761 (2010).

5. *Finally, the authors are in an optimal position to address if the genetic components of top MRF detected for LIFO are also associated with the LIFO phenotype. This is especially relevant for diabetes because an enormous controversy exists about the genetic connection between Alzheimer/Neurodegenerative diseases and diabetes risk susceptibility loci (as an example see a*

very recent report by Escott Price et al. [https://doi.org/10.1016/s1474-4422\(22\)00395-7](https://doi.org/10.1016/s1474-4422(22)00395-7) and references therein).

Again, thank you for this suggestion. We took the top 3 MRFs: 1. diabetes: “diabetes diagnosed by doctor”, 2. pollution: “nitrogen dioxide air pollution”, and 3. alcohol: “alcohol intake frequency”, searched for UK Biobank GWAS results on these 3 variables (available from the Open Targets Genetics website: <https://genetics.opentargets.org/>), and compared the significant hits within ± 250 kbp of, or in high LD (>0.8) with, our own LIFO loci. When hits were limited (such as for “nitrogen dioxide air pollution”, predictably), we also searched online for GWAS done on similar variables. Finally, we also included the list of significant hits for diabetes from the reference given above by Reviewer 1. We **found some genetic overlap for alcohol intake frequency**, with two hits belonging to the 2nd (chromosome 4) and 5th (chromosome 17) clusters, respectively.

In addition, we have now also **highlighted and updated in Table 1 the significant associations of our genetic hits with UK Biobank variables related to the 15 categories** listed for the MRFs, as well as schizophrenia and Alzheimer’s disease (and cognitive performance/memory). Namely, the 1st cluster is related to alcohol intake, inflammation, diabetes, food intake, as well as Alzheimer’s disease; the 2nd cluster with cholesterol, blood pressure, weight, diabetes, inflammation, alcohol, and sleep (plus cognitive performance/memory); the 3rd cluster with weight; the 4th with diabetes, as well as schizophrenia; and the 5th with alcohol and sleep.

We have now **described these additional analyses in a dedicated section of the Methods:**

Methods: *Associations between the LIFO brain network’s genetic hits and the MRFs.* First, we reported in **Table 1** the significant associations between the LIFO genetic hits and UK Biobank variables related to the 15 categories listed for the MRFs. For this, we used the Open Targets Genetics website, which reports the GWAS carried out in the UK Biobank (<https://genetics.opentargets.org/>). Second, we assessed whether there was any genetic overlap between the known genetic components of the best 3 MRFs and the LIFO phenotype. Again, we used the Open Targets Genetics website outputs for these 3 very specific UK Biobank variables, and compared the significant hits for these best 3 MRFs within ± 250 kbp of, or in high LD (>0.8) with, our own LIFO variants. If reported hits were limited, we also searched online for GWAS done on similar variables. Finally, we also included the list of significant hits for diabetes⁵⁵, which focused on a potential genetic overlap between diabetes and Alzheimer’s disease.

And have **re-written and expanded parts of the Results and Discussion to highlight these associations:**

Results: The first autosomal genetic cluster, on chromosome 1, included two variants (lead variant: rs6540873, $\beta = 0.06$, $P = 1.71 \times 10^{-8}$, and rs1452628, with posterior probabilities of inclusion in the causal variant set of 0.56 and 0.45, respectively) close to, and eQTL of, *KCNK2 (TREK1)*. This gene regulates immune-cell trafficking into the central nervous system, controls inflammation, and plays a major role in the neuroprotection against ischemia. Of relevance, these two loci are in particular related in UK Biobank participants with the amount of alcohol consumed, insulin levels, inflammation with interleukin-8 levels, as well as, crucially, with late-onset Alzheimer's disease (**Table 1**).

The second autosomal genetic cluster on chromosome 4 is made of 7 loci, with the lead variant rs13107325 in an exon of *SLC39A8/ZIP8* ($\beta = 0.14$, $P = 2.82 \times 10^{-13}$, posterior probability: 0.99). This locus is one of the most pleiotropic SNPs identified in GWAS, and is, amongst many other associations, related in UK Biobank with cholesterol, blood pressure, weight, inflammation with C-reactive proteins levels, diabetes with insulin-like growth factor 1 levels, alcohol intake, sleep duration, and cognitive performance/impairment, including prospective memory.

The third locus was an indel in chromosome 6 in an intron, and eQTL, of *RUNX2* (rs35187443, $\beta = 0.06$, $P = 9.03 \times 10^{-9}$), which plays a key role in differentiating osteoblasts, and has been shown very recently to limit neurogenesis and oligodendrogenesis in a cellular model of Alzheimer's disease¹⁰.

The fourth locus was a SNP in chromosome 12, in an intron of *NUAK1* (rs12146713, $\beta = -0.10$, $P = 1.26 \times 10^{-9}$), and remarkably its top association in the UK Biobank was with the contrast between schizophrenia and major depressive disorder¹¹, and it was also associated with insulin-like growth factor 1 levels.

The final genetic autosomal genetic cluster was made of 3,906 variants in the *MAPT* region. Its lead non-triallelic variant, rs2532395 ($\beta = -0.09$, $P = 3.56 \times 10^{-15}$) was more specifically <10kb from *KANSL1* and an eQTL of *KANSL1*, *MAPT* and other genes in brain tissues. This locus was also associated in UK Biobank with tiredness and alcohol intake. *MAPT* is in 17q21.31, a chromosomal band involved with a common chromosome 17 inversion¹². Adding chromosome 17 inversion status as a confounder reduced the significance of the association ($\beta = -0.15$, $P = 8.45 \times 10^{-3}$). (...)

(...) In particular, amongst the more easily interpretable findings of the most associated variables with rs312238, the T allele of this locus was associated with two increased measures of deprivation and/or disability (worse socioeconomic status), the 'Townsend deprivation index' and the 'Health score', but also with 'Nitrogen dioxide air pollution', 'Maternal smoking around birth', as well as 'Number of full brothers' and 'Number of full sisters', thus showing consistent signs of association between this variant and those phenotypes.

(...)

While 6 out of the 7 genetic clusters associated with the LIFO network were correlated with many variables related to each of the 15 MRF categories, including diabetes, alcohol consumption and traffic pollution (**Table 1**), we also found some genetic overlap between the very specific best MRF of "alcohol intake frequency" and the LIFO network in the pleiotropic rs13107325 variant (cluster 2), as well as rs17690703, part of the large genetic cluster 5 in *MAPT* (**Table S3**). No genetic overlap was found for the precise "nitrogen dioxide air pollution in 2005" or "diabetes diagnosed by doctor", nor for more approximate variables.

Discussion: All five autosomal genetic clusters identified through the GWAS on the LIFO phenotype had relevant associations with risk factors for dementia (Results; **Table 1**), including precisely two of the best MRFs (clusters 2 and 5), and three of them directly related in UK Biobank to the two diseases

showing a pattern of brain abnormalities following the LIFO network: schizophrenia (clusters 2 and 4) and Alzheimer's disease (cluster 1)(Table 1).

(...)

In line with the LIFO brain network being both prone to accelerated ageing and susceptible to Alzheimer's disease, this genetic locus has also been associated genome-wide with well-known risk factors for dementia. These include alcohol – including the exact same variable of “alcohol intake frequency” as identified as one of the best MRFs – cholesterol, weight, sleep – including “sleep duration” – and blood pressure²³⁻²⁷, all of which significantly contribute to modulating the LIFO brain network when considered separately (Table 2).

References: 10. Nakatsu, D. et al. BMP4-SMAD1/5/9-RUNX2 pathway activation inhibits neurogenesis and oligodendrogenesis in Alzheimer's patients' iPSCs in senescence-related conditions. Stem Cell Reports 18, 1246, doi:10.1016/j.stemcr.2023.03.017 (2023).

11. Peyrot, W. J. & Price, A. L. Identifying loci with different allele frequencies among cases of eight psychiatric disorders using CC-GWAS. Nat Genet 53, 445-454, doi:10.1038/s41588-021-00787-1 (2021).

12. Steinberg, K. M. et al. Structural diversity and African origin of the 17q21.31 inversion polymorphism. Nat Genet 44, 872-880, doi:10.1038/ng.2335 (2012).

55. Hardy, J., de Strooper, B. & Escott-Price, V. Diabetes and Alzheimer's disease: shared genetic susceptibility? Lancet Neurol 21, 962-964, doi:10.1016/S1474-4422(22)00395-7 (2022).

Reviewer #2:

This manuscript presents an interesting research question of clinical relevance: the identification of common alleles and 'modifiable' factors influencing the neuroanatomy of a group of brain regions shown to have an age-related slower development, but a quicker degeneration (named here 'Late In, First Out' [LIFO]). This group of brain regions have also shown spatial correlation with changes seen in participants with Alzheimer and Schizophrenia; suggesting being a 'vulnerable' set of regions for neurodevelopmental/neurodegenerative disorders. Consequently, the authors run to separate analyses: A GWAS of averaged volume of the brain regions within this network, and a series of GLM analyses to examine the association between different 'modifiable' factors and the same brain phenotype. Despite the methods applied are sound, I feel that the analyses conducted do not fully address the question posed and they only scratch the surface of the study's aim. As such, I have doubts about the real significance and impact of the results reported.

We are pleased that Reviewer 2 highlighted the soundness of our methodological approaches, and we address below their comments.

1. *I understand that the authors base their approach on a previous paper where they identified the LIFO network, and that this was done based on regional brain volumes. Consequently, they calculate a weighted average grey matter volume measure for the LIFO. However, from a genetic*

perspective, cortical volume is a less interesting measure – at least to me – than their components of cortical thickness and surface area. We know that these two cortical metrics are under different genetic influences, showing little genetic correlation (Grasby et al, Science 2020). It would be interesting to realise whether cortical thickness and/or surface area in this brain network present with quadratic developmental trend, in other words, if any of the two is a major contributor to this network characteristic trend. Then, a GWAS of the selected metric, or two separate GWASes for each of them if both show to contribute, would have increased interest to me. Moreover, I would be interested in knowing how the results compare to those of Grasby et al., who conducted GWAS not only for overall thickness and surface area, but also for all cortical gyri based on the Desikan-Killiany atlas separately.

The uniqueness of this study relies on the fact that we **combined the strengths of two different cohorts**: the first, which revealed the LIFO network, is lifespan, demonstrating the mirroring of developmental and ageing processes in the LIFO brain areas, something that can **never be achieved with UK Biobank** that has a limited age range. For this initial work with the lifespan cohort (Douaud *et al.*, PNAS 2014), we not only included grey matter (GM) partial volume images, as done in this current study, but also Freesurfer information of cortical thickness and cortical area. A lot of the GM networks emerging from the linked independent component analysis carried out on these three types of structural images (GM partial volume, Freesurfer cortical thickness, and Freesurfer surface area) revealed shared variance and common spatial distribution between cortical area and GM volume (something we demonstrated directly in Smith *et al.*, J Neurosci 2019). However, the LIFO network – one of only two brain components to show a *post hoc* relationship with age – **showed no contribution from Freesurfer cortical area or thickness**. This might hint at processes that only partial volume maps are able to detect due to its specific localisation, which spans regions that include the cerebellum and subcortical structures, which are not included in the area and thickness surface methods from Freesurfer.

Since the main objective of this work is to study the modifiable and non-modifiable (genetics) factors playing a role in modulating this very specific LIFO brain network, precisely because of its distinctive characteristics, we strongly believe that our imaging approach is the best to address our question of interest. We have now **added a paragraph in the Discussion** expanding on this point:

Discussion: The uniqueness of this study relies on the fact that we combined the strengths of two different cohorts: the first, which revealed the LIFO grey matter network, is lifespan, demonstrating the mirroring of developmental and ageing processes in the LIFO brain areas. This could never be achieved with UK Biobank because of its limited age range. Of note, for this initial work with the lifespan cohort⁸, we not only included grey matter partial volume images, as done in this current study, but also Freesurfer information of cortical thickness and surface area. The LIFO network

showed no contribution from Freesurfer cortical thickness or area. This might hint at processes that only partial volume maps are able to detect due to the LIFO network's specific localisation, including in the cerebellum and subcortical structures, which are not included in the area and thickness surface methods from Freesurfer.

2. Furthermore, *postGWAS* basic analyses (e.g. *finemapping*) would add interest to the results,

We thank Reviewer 2 for this suggestion. We agree that fine mapping deepens our understanding of the results and to this purpose, we had provided LocusZoom plots for the 5 autosomal genetic clusters. Since an update of the online tool makes this now possible, we have also **created the LocusZoom plots for the two genetic clusters in PAR1** of the sex chromosomes:

Figure 2: (...) Bottom row, regional association plots of the top variants for the two genetic clusters in the pseudo-autosomal region PAR1 of the X chromosome: rs312238 (XG, CD99) and rs2857316 (XG)(UK Biobank has no genotyped variants on the 3' side).

In addition to doing the mediation analyses between LIFO genetic hit in *MAPT* and Alzheimer's disease (see Reviewer 1, Point 4.) and associations of LIFO genetic clusters and the modifiable risk factors for dementia (Reviewer 1, Point 5.), we now also **carry out two additional post-GWAS analyses**: 1. we assess **causality among the significant variants** for each autosomal genetic cluster using CAVIAR; and 2. we use PANTHER for **enrichment analysis** of the genes involved with our significant loci. We have **added the corresponding following text to the Methods section**:

Methods: *Causality within each genetic cluster.* We used CAVIAR (Causal Variants Identification in Associated Regions⁴⁹) to assess causality of variants that passed the genome-wide significance threshold in each of the genetic clusters we report. CAVIAR uses a Bayesian model and the local linkage disequilibrium structure to assign posterior probabilities of causality to each variant in a region, given summary statistics for an association. We did not perform CAVIAR analysis on the genetic cluster on chromosome 17, as its lead variant (rs2532395) was strongly correlated with chromosome 17 inversion, and the LD matrix was large and low rank. We excluded the X chromosome loci from this analysis due to the difficulty in assessing LD in this chromosome.

Enrichment analysis. Based on the genes listed in the 'Genes' column of Table 1, we performed an enrichment analysis for the genes associated with the LIFO brain network using PANTHER⁵⁰. PANTHER determines whether a gene function is overrepresented in a set of genes, according to the gene ontology consortium. We also considered another version of this analysis in which the genes of the *MAPT* extended region are removed from the input to PANTHER^{51,52}.

And we have **included the information provided by the causality analysis as part of our much expanded genetic section of the Results** (see also Reviewer 1, Point 5.):

Results: The first autosomal genetic cluster, on chromosome 1, included two variants (lead variant: rs6540873, $\beta = 0.06$, $P = 1.71 \times 10^{-8}$, and rs1452628, with posterior probabilities of inclusion in the causal variant set of 0.56 and 0.45, respectively) close to, and eQTL of, *KCNK2 (TREK1)*. This gene regulates immune-cell trafficking into the central nervous system, controls inflammation, and plays a major role in the neuroprotection against ischemia. Of relevance, these two loci are in particular related in UK Biobank participants with the amount of alcohol consumed, insulin levels, inflammation with interleukin-8 levels, as well as, crucially, with late-onset Alzheimer's disease (**Table 1**).

The second autosomal genetic cluster on chromosome 4 is made of 7 loci, with the lead variant rs13107325 in an exon of *SLC39A8/ZIP8* ($\beta = 0.14$, $P = 2.82 \times 10^{-13}$, posterior probability: 0.99). This locus is one of the most pleiotropic SNPs identified in GWAS, and is, amongst many other associations, related in UK Biobank with cholesterol, blood pressure, weight, inflammation with C-reactive proteins levels, diabetes with insulin-like growth factor 1 levels, alcohol intake, sleep duration, and cognitive performance/impairment, including prospective memory.

(...)

The top variants for these clusters were related to two homologous genes coding for the two antigens of the XG blood group: rs312238 ($\beta = -0.05$, $P = 1.77 \times 10^{-10}$, with a posterior probability of inclusion in the causal variant set of 0.99) ~10kb from, and an eQTL of, *CD99/MIC2*, and rs2857316 ($\beta = -0.08$, $P = 2.27 \times 10^{-29}$) in an intron and eQTL of *XG*.

Finally, we **discuss the findings from the gene enrichment analysis**:

Discussion: Intriguingly, an analysis revealed that the genes involved in the loci associated with the LIFO network (**Table 1**) are enriched for the gene ontology terms of leucocyte extravasation, namely “positive regulation of neutrophil extravasation” ($P = 4.75 \times 10^{-6}$) and “T cell extravasation” ($P = 4.75 \times 10^{-6}$). This result held when removing the genes included in the *MAPT* extended region (with $P = 2.54 \times 10^{-6}$ and $P = 2.54 \times 10^{-6}$, respectively). Leucocyte extravasation facilitates the immune and inflammatory response, and there has been renewed focus on the fact that a breakdown of the blood-brain barrier together with leukocyte extravasation might contribute to both Alzheimer’s disease and schizophrenia^{33,34}. In line with the enrichment finding, 4 out of the 7 genetic clusters associated with the LIFO network are correlated in UK Biobanks’ blood assays with percentage or count of immune cells (neutrophil, lymphocyte, platelet, monocyte, etc.; **Table 1**).

References: 33. Zenaro, E., Piacentino, G. & Constantin, G. The blood-brain barrier in Alzheimer’s disease. *Neurobiology of disease* 107, 41-56, doi:10.1016/j.nbd.2016.07.007 (2017).

34. Pong, S., Karmacharya, R., Sofman, M., Bishop, J. R. & Lizano, P. The Role of Brain Microvascular Endothelial Cell and Blood-Brain Barrier Dysfunction in Schizophrenia. *Complex Psychiatry* 6, 30-46, doi:10.1159/000511552 (2020).

50. Hormozdiari, F., Kostem, E., Kang, E. Y., Pasaniuc, B. & Eskin, E. Identifying causal variants at loci with multiple signals of association. *Genetics* 198, 497-508, doi:10.1534/genetics.114.167908 (2014).

51. Thomas, P. D. et al. PANTHER: Making genome-scale phylogenetics accessible to all. *Protein Sci* 31, 8-22, doi:10.1002/pro.4218 (2022).

52. Ashburner, M. et al. Gene ontology: tool for the unification of biology. The Gene Ontology Consortium. *Nat Genet* 25, 25-29, doi:10.1038/75556 (2000).

53. Gene Ontology, C. et al. The Gene Ontology knowledgebase in 2023. *Genetics* 224, doi:10.1093/genetics/iyad031 (2023).

3. *as well as the investigation of genetic correlations with other phenotypes (particularly Alzheimer’s disease and schizophrenia). In my opinion, this section of the manuscript, taken to a far deeper investigation than currently presented, would represent in itself a very interesting contribution to the field.*

Thank you for this suggestion. Following Reviewer 1’s comments, we have now looked at genetic overlap between the LIFO and the 3 best MRFs (alcohol, diabetes and pollution), associations in UK Biobank between the genetic hits and relevant phenotypes (in the updated **Table 1**), and at the mediation between our genetic top hit in *MAPT* and Alzheimer’s disease via the LIFO brain network. We have found that the

lead bi-allelic variant in *MAPT* has an effect on Alzheimer's disease diagnosis via the LIFO brain regions. We have also identified that three of the seven genetic clusters are *directly* associated with schizophrenia or Alzheimer's disease in UK Biobank:

Results: All five autosomal genetic clusters identified through the GWAS on the LIFO phenotype had relevant associations with risk factors for dementia (Results; **Table 1**), including precisely two of the best MRFs (clusters 2 and 5), and three of them directly related in UK Biobank to the two diseases showing a pattern of brain abnormalities following the LIFO network: schizophrenia (clusters 2 and 4) and Alzheimer's disease (cluster 1)(**Table 1**).

Furthermore, we have now **carried out an analysis of the shared heritability** between LIFO and Alzheimer's, and between LIFO and schizophrenia, and **detail it in the Methods:**

Methods: Heritability. We examined the heritability of the LIFO phenotype, and the coheritability between the LIFO network and Alzheimer's disease or schizophrenia using LDSC⁵⁷. This method uses regression on summary statistics to determine narrow sense heritability h^2 of a trait, or the shared genetic architecture between two traits. LDSC corrects for bias LD structure using LD calculated from a reference panel (we used LD from the Thousand Genomes Project Phase 1⁵⁸). We obtained summary statistics for a meta-analysis of Alzheimer's disease involving 71,880 cases and 383,378 controls⁵⁹. The number of genetic variants in the intersection between the summary statistics was 1,122,435. For schizophrenia, the summary statistics were obtained from a meta-analysis involving 53,386 cases and 77,258 controls⁶⁰. A total of 1,171,319 genetic variants were in the intersection with the summary statistics for LIFO. For both Alzheimer's and schizophrenia, the X chromosome was not included in the heritability calculation, as it was excluded from the meta-analysis that we sourced the summary statistics from.

And report it in the Results:

Results: The genetic co-heritability between the LIFO network and Alzheimer's disease or schizophrenia was not statistically significant (coefficient of co-heritability = -0.12, se = 0.10; $P=0.23$; coefficient of co-heritability = -0.16, se = 0.04, $P = 0.07$, respectively).

References: 59. Jansen, I. E. et al. Author Correction: Genome-wide meta-analysis identifies new loci and functional pathways influencing Alzheimer's disease risk. *Nat Genet* 52, 354, doi:10.1038/s41588-019-0573-x (2020).

60. Trubetskoy, V. et al. Mapping genomic loci implicates genes and synaptic biology in schizophrenia. *Nature* 604, 502-508, doi:10.1038/s41586-022-04434-5 (2022).

4. *Instead of deepening their GWAS analysis, the authors present a separated analysis on the association between 'modifiable' risk factors and LIFO's weighted average grey matter volume. I do have reservations about the interest of this analysis and how it is conducted. It does not appear clear to me how the 15 categories in which variables are grouped are defined, neither how*

each variable is assigned to them? Why did the authors not enter in their regression all variables with missingness above their selected criteria? Only taking one variable per category based on the lowest p value does not appear the best criteria to me, considering that all association values are very small and in most cases with neglectable differences among them, and since these are models with a single predictor of interest, the significant variables discarded might have been explaining separate variance from the one selected?

We have now **deepened considerably our GWAS analysis** by doing fine mapping, adding the LocusZoom plots of the significant variants on the X chromosome, looking at heritability of our phenotype of interest and its shared heritability with schizophrenia and Alzheimer's disease, pursuing a mediation analysis for chromosome 17, carrying out an enrichment analysis, and by looking at associations between our genetic hits and the 15 categories of modifiable risk factors.

These 15 categories were chosen as some of the most reported modifiable risk factors in unhealthy ageing. Out of these 15, **11 are in common with the recent updated Lancet commission on dementia prevention**, and we added the cerebrovascular factor of cholesterol, as well as the well-known risk factors of sleep, diet, and inflammation (Livingston *et al.*, 2020).

Here, we wanted to **optimise the interpretability of the results**, and to be able to relate them to previous findings. Entering all variables passing the missingness threshold, for instance by taking the first PC for each category, would have prevented us from being able to report exactly which variable – and subsequently, which genetic component for this specific variable – contribute to the effect. For these same reasons, we did not create any compound variable either, such as total cholesterol: LDL ratio.

Moreover, low missingness is absolutely essential not to induce biases and lose too many participants. While 5% of missingness seems like a stringent threshold, these missing data are from different, non-overlapping participants, as witnessed by our final sub-sample being reduced by more than 10% (from 39,676 to 35,527 participants). To be able to perform a data reduction such as a PCA, and do so consistently for all the MRF categories, we would need to consider the *complete* data for each category, which, in practice, would have limited considerably the number of variables taken into account for the PCA. Further, this would have **greatly reduced the number of participants** investigated in our model combining all categories at once (for which complete data are again necessary), likely **creating a bias in the sub-sampled population** used for the final, comprehensive model. This is all the more probable given that **data are not missing at random in UK Biobank**, and this missingness has some genetic component (Mignogna *et al.*, 2023).

We have now **expanded the justification** for these choices made in the Methods section:

Methods: The following 15 categories of modifiable risk factors (MRFs) for dementia were investigated based on previous literature: blood pressure, diabetes, cholesterol, weight, alcohol, smoking, depression, hearing, inflammation, pollution, sleep, exercise, diet/supplementation,

socialisation, and education. These included well-documented cerebrovascular risk factors, and in particular included all of the 12 modifiable risk factors considered in the updated Lancet commission on dementia, with the sole exception of traumatic brain injury³. For each category, several MRF variables from the UK Biobank were very minimally pre-processed (**Supplementary Information**). In total, 161 MRF variables were obtained. To optimise the interpretability of the results, and to be able to relate them to previous findings, we did not carry out any data reduction, which would have prevented us from identifying exactly which variable – and subsequently, which genetic component for this specific variable – contribute to the effect. For these same reasons, we did not create any compound variable.

(...)

A prerequisite to carry out this single general linear model analysis was to only include participants who would have values for all best MRFs and confounders. This explains the additional criterion of only including MRFs that had no more than 5% of values missing, to ensure that the final sample of participants who had values for all these best and confounding factors would not be biased compared with the original sample – something we formally tested (see **Supplementary Information**) – especially as data are not missing at random in UK Biobank, and exhibit some genetic structure⁴⁸.

References: 3. Livingston, G. et al. Dementia prevention, intervention, and care: 2020 report of the Lancet Commission. *Lancet* 396, 413-446, doi:10.1016/S0140-6736(20)30367-6 (2020).

49. Mignogna, G. et al. Patterns of item nonresponse behaviour to survey questionnaires are systematic and associated with genetic loci. *Nat Hum Behav*, doi:10.1038/s41562-023-01632-7 (2023).

5. *Some of these ‘modifiable’ factors are under a significant genetic influence and might have been simply more interesting to determine to what extent there’s genetic overlap between common alleles influencing these and the GWAS results for LIFO’s weighted average volume? ... I do not really understand this second part of the manuscript and how it all fits with the GWAS analysis done previously.*

The starting point, and our main interest for this work, revolves around what might contribute to the modulation of the LIFO brain network, because of its implication in (abnormal) development and ageing, and its spatial correspondence with schizophrenia and Alzheimer's disease brain patterns. These factors might be modifiable, or not modifiable (genetics); hence the two approaches, which give complementary information, presented in our study.

We agree with Reviewer 2 however that integrating further both sets of results would be beneficial. We had already established that one of our two sex chromosome hits was significantly associated with air pollution. Following Reviewer 1's Point 5., we have now determined the genetic overlap between the 3 best MRFs and the significant variants found for the LIFO brain regions, and looked at associations in UK Biobank between the LIFO genetic hits and variables from the same 15 MRF categories.

6. *Another aspect that surprised me of this manuscript was the fact that only sex, age and brain volume were added as covariates in the analyses. However, the authors have demonstrated the importance of accounting for many other confounders in UK Biobank’s imaging data such are*

head movement, head position, table position, assessment centre, etc. (Alfaro-Almagro et al, 2021 Neuroimage).

Reviewer 2 is absolutely right. After discussing with the lead author of the “confounding variables” paper mentioned above (Alfaro-Almagro et al., 2021), we had in fact **investigated the effect of 32 possible confounders** tailored for structural imaging, including all of those mentioned by Reviewer 2. They also had the added benefit of having complete data. Demographics of age, quadratic age, sex, age × sex and quadratic age × sex, as well as head size had a clear effect on the LIFO values, so we retained those 6 confounders. We have now made this **explicit in the Methods**, and **provided a new Supplementary Table**:

Methods: In total, 32 variables tailored to structural imaging had been considered as possible confounders, and we retained those with the strongest association ($R^2 \geq 0.01$; **Table S5**).

Reviewer #3:

The present manuscript reports the genetic and phenotypic correlates of the previously-characterised LIFO brain network (grey matter loci whose grey matter is most strongly correlated with cross-sectional age differences using a VBM-based ICA) in the UK Biobank imaging sample. Associations with this network across the genome (GWAS) and across a large number of raw modifiable risk factors (MRFs) are reported, with relatively few, but plausible, hits of small effect size across both sets of results. I like very much that the LIFO network as a quantitative response to the retrogenesis hypothesis/observations, and it is one valuable and important approach for distilling out those structural properties which are most interesting to subject to the types of testing presented here. These are reasonable questions to ask, and the manuscript has several strengths including LIFO having been trained out of sample, the desire to seek GWAS replication (within two UKB subsets), and an interest in multivariable analyses to identify unique contributions of MRFs, though there are a number of analytical and interpretative issues which I raise below.

Thank you very much to Reviewer 3 for appreciating our work around the retrogenesis, and for acknowledging the efforts we have put in the analyses.

1. *Sincere apologies if I have missed it, but I could not see that the authors had taken steps to control for technical covariates in their genetic analyses. Typically batch, array and PCs for population stratification.*

Our mistake entirely, this was somehow not explicitly stated in our Methods when this should have. For all the genetic associations reported in this work, we **used 40 genetic principal components** as confounders. This principal component confound set is provided by UKB, and is the one typically used for brain imaging work on UKB. We used no other confounds than these 40 PCs in our previous GWAS studies (e.g. Elliott et al., 2018; Smith et al., 2021). In particular, we did not use batch or array components as confounds. Indeed, as only a subset of UKB participants are imaged, the number of

unique batch identifiers is quite large compared to the number of subjects, which would lead to singularities in the linear test should they be included as confounders. We have **added the following text to the Methods section:**

Methods: To account for population stratification, 40 genetic principal components were used in the genetic association tests as is recommended for UK Biobank genetic studies^{20,21,48}.

References: 20. Elliott, L. T. et al. Genome-wide association studies of brain imaging phenotypes in UK Biobank. *Nature* 562, 210-216, doi:10.1038/s41586-018-0571-7 (2018).

21. Smith, S. M. et al. An expanded set of genome-wide association studies of brain imaging phenotypes in UK Biobank. *Nat Neurosci* 24, 737-745, doi:10.1038/s41593-021-00826-4 (2021).

48. Bycroft, C. et al. The UK Biobank resource with deep phenotyping and genomic data. *Nature* 562, 203-209, doi:10.1038/s41586-018-0579-z (2018).

2. *GWAS analyses – can the authors provide further information on why they opted to discard data/power on related individuals when it is possible to preserve power and model kinship?*

We thank the reviewer for this suggestion – it is true that using a model such as a linear mixed model would allow us to follow test assumptions while including related individuals. However, this would lead to recovery of only around 450 samples, and would have a negligible impact on power, or our results considering their strengths. We now **specify this in the Methods:**

Methods: Briefly, we examined imputed UK Biobank genotype data⁴⁸, and restricted the analysis to samples that were unrelated (thereby setting aside only ~450 participants), without aneuploidy and with recent UK ancestry.

3. *There doesn't seem to be any acknowledgement that cross-sectional data were used as a proxy for longitudinal ageing effects (both in training and testing). Is it known how well the LIFO network data can be reconciled with within-person trends?*

We agree entirely with Reviewer 3 that cross-sectional data cannot make up for having access to rates of change via longitudinal studies. We now **acknowledge this as a clear limitation of our study**, including references of work showing the occasional disconnect between the two:

Discussion: It is also important to note that cross-sectional and longitudinal patterns of brain ageing can differ, as has been shown for instance for episodic and semantic memory adult span trajectories, especially in younger adults³⁹. A recent study has also demonstrated a specific 'brain age' imaging measure to be more related to early life influences on brain structure than within-person rates of change in the ageing brain⁴⁰. Further work will be needed to establish how the LIFO network changes in terms of within-person trends, for instance by investigating the growing UK Biobank longitudinal imaging database.

References: 39. Ronnlund, M., Nyberg, L., Backman, L. & Nilsson, L. G. Stability, growth, and decline in adult life span development of declarative memory: cross-sectional and longitudinal data from a population-based study. *Psychol Aging* 20, 3-18, doi:10.1037/0882-7974.20.1.3 (2005).

40. Vidal-Pineiro, D. et al. Individual variations in 'brain age' relate to early-life factors more than to longitudinal brain change. *Elife* 10, doi:10.7554/eLife.69995 (2021).

4. *The LIFO was trained on a smaller cross-sectional dataset some time ago. I recognise the value in having lifecourse developmental MRI data, not available in UKB. However, with the much larger UKB sample in mind, to what extent are the near-absent outer-cortical loci in the LIFO mask due to gyral folding/statistical power differences across the brain space in the initial discovery cohort? Would it boost power/precision for the GWAS and RF analyses to use a UKB discovery sample to re-run the VBM ICA? Whether it yields fundamentally different loci is an empirical question. In fact, are areas which are ageing-related but not important for development of additional interest (though of course, this later analyses might be beyond the scope of the present paper)?*

Thank you – we agree that these would be interesting questions to explore, but these are truly beyond the scope of this particular study that leverages the lifespan aspect of the original dataset, and as such, rests entirely on the 70 brain maps drawn from it.

5. *The authors report some genetic clusters whose associations with brain measures have been reported previously. Similarly, many of the associations between the top MRFs and brain measures have been identified previously both in UKB and other samples. Whereas it is good to have replicated those findings, to what extent is the LIFO weighted mask just telling us about overall grey matter ~ genotype/phenotype associations? That is, can the authors draw out more clearly (in empirical terms) for the reader that LIFO is getting at something statistically more informative – with respect to gene and RF - than one could identify examining all GM, or variants thereof (such as sulcal widening or brain age) which have previously reported some GWAS hits reported here?*

Reviewer 3 is right, some of the genetic clusters were indeed found in other brain imaging studies, including one of 'brain age' and one looking at global sulcal widening, suggesting that these are related to gross markers of ageing in the brain, rather than anything more specific. Having said that, this is only true for 4 out of 7 of genetic clusters (in the larger sense, since cluster 5 that comprises 3,906 significant variants in the *MAPT* extended region only includes two loci overlapping with Smith et al., 2021). Our recent study (Smith et al., 2021), expanding on our GWAS of all brain IDPs (Elliott et al., 2018), not only included the same genotyped and scanned UKB participants as in this current work, but also included the X chromosome. Importantly, despite the fact that it encompassed all of the brain IDPs, **it did not uncover three of our genetic clusters associated with the LIFO network**. This suggests that **these three clusters are specific to our phenotype of interest** here. Moreover, **two of these clusters are in PAR1 of the sex chromosomes, which is an entirely novel finding**, and points for the first time at a possible role of the XG blood group in ageing (or development) in the brain.

On the other hand, regarding the MRFs analyses, our intent was precisely to focus on the most commonly known modifiable risk factors for dementia, and these will have

unsurprisingly been found associated with brain imaging markers. Nevertheless, we note that **traffic-related pollution has only been relatively recently shown to be associated with brain alterations**. The novelty of our approach also resides, as highlighted by Reviewer 3, in combining the risk factors in a single model to tease out their unique contributions.

We have now **updated Table 1 to include an additional column** summarising if our current genetic findings have been previously identified in our expanded GWAS study. We have also **written a dedicated paragraph in the Discussion** emphasising the novelty of the genetic results:

Discussion: On the other hand, of the seven genetic clusters, three were entirely novel (clusters 3, 6 and 7), and not found in other brain imaging studies, including our most recent work that expanded on our previous GWAS of *all of the brain IDPs available* in UK Biobank²⁰ by including more participants – in fact, the same number of participants as analysed in this present work – and, crucially, by also including the X chromosome²¹ (**Table 1**). This suggests that, beyond the genetic hits that were meaningfully associated with the LIFO brain network and an array of relevant risk factors, lifestyle variables and brain disorders, and found in a few other imaging GWAS, some of the genetic underpinnings of the LIFO network are intrinsically specific to it and to no other pre-existing imaging phenotype.

References: 20. Elliott, L. T. et al. Genome-wide association studies of brain imaging phenotypes in UK Biobank. *Nature* 562, 210-216, doi:10.1038/s41586-018-0571-7 (2018).

21. Smith, S. M. et al. An expanded set of genome-wide association studies of brain imaging phenotypes in UK Biobank. *Nat Neurosci* 24, 737-745, doi:10.1038/s41593-021-00826-4 (2021).

6. *Modifiable RF study: Given several co-authors of the current study conducted a detailed study highlighting the effect of modelling imaging confounds in UK Biobank, the use of a very restricted set in the current analyses – just age and sex (++) and head size (it that skull scaling?) – could do with a brief additional comment.*

Thanks - we agree with this and should have been more explicit in our strategy to select the confounders (see also Reviewer 2's Point 6.). We in fact had explored 32 confounders tailored to structural imaging and that also had the advantage of presenting with complete data. Of these, 6 had a clear impact, and we used them in the final modelling. We have now **clarified this in the Methods**, and **added the corresponding Supplementary Table**:

Methods: In total, 32 variables tailored to structural imaging had been considered as possible confounders, and we retained those with the strongest association ($R^2 \geq 0.01$; **Table S5**).

7. *State briefly in the text whether the directions of associations between rs312238 and the mentioned phenotypes are in a consistent/expected direction.*

Thank you for this suggestion. We have now examined the directions of associations between rs312238 and the list of non-imaging derived phenotypes (nIDPs) listed in **Table S2**. Broadly, this variant is **consistently associated with socioeconomic, environmental, and health indicators**. In particular, amongst the most easily

interpretable findings, the T allele of rs312238 is associated with an increased Townsend deprivation index (worse socioeconomic status). Aside from Townsend index, the top 35 most associated nIDPs also include the direct socioeconomic indicator 'Health score' – an index of health deprivation and disability – with the T allele associated with an increase. The direction of association for the direct measures of deprivation among the top nIDPs thus match. In addition, the measure of environmental exposure corresponding to our “best” MRF of pollution ('Nitrogen dioxide air pollution'), and other specific measures of early life factors/deprivation (such as 'Maternal smoking around birth', 'Number of full brothers' and 'Number of full sisters') are also positively associated with the T allele and have matching signs. We now **comment on the direction of these associations in the Results section:**

Results: In particular, amongst the more easily interpretable findings of the most associated variables with rs312238, the T allele of this locus was associated with two increased measures of deprivation and/or disability (worse socioeconomic status), the 'Townsend deprivation index' and the 'Health score', but also with 'Nitrogen dioxide air pollution', 'Maternal smoking around birth', as well as 'Number of full brothers' and 'Number of full sisters', thus showing consistent signs of association between this variant and those phenotypes.

8. *Suggest more careful phrasing of causal language throughout – e.g. “modifiable risk factors’ influences over the vulnerable LIFO brain network”*

Absolutely – we have now combed the manuscript for words such as “influence” and “impact”, and **removed them accordingly.**

9. *The title seems to refer to the LIFO as being *the* brain network most vulnerable to disease. I am not sure that has been empirically demonstrated, has it? On a related note, does the absence of APOE associations here – a well-known genetic risk factor for brain and cognitive ageing, and dementia – potentially speak to the lack of statistical power for the GWAS setup (which may have been reduced by opting for a discovery/replication sample rather than a single-sample analysis)?*

“Most” vulnerable in the title actually related to “ageing”, which is empirically what we have established for this brain network out of the 70 networks described in our PNAS 2014/J Neurosci 2019. Nevertheless, we have now **removed “most” to soften the title,** especially as we cannot be more specific due to title length constraints.

There does not seem to be **any variant even close to significance in chromosome 19**, so this may suggest that either *APOE* does not target the same regions as defined by the LIFO network (and indeed, even if the hippocampus appears in both, the LIFO network is quite distinct from the DMN), or not in a way that gets specifically captured by our FLICA approach.

REVIEWERS' COMMENTS

Reviewer #2 (Remarks to the Author):

The authors have now run a more indepth post-gwas analysis of their results that in my view has improved the manuscript and increased the interest of their results. They have also clarified their strategy to select MRFs and what really these represented. All my concerns have been adequately addressed and I have not further queries.

Reviewer #3 (Remarks to the Author):

The authors have addressed all my comments satisfactorily.